# Beta traveling waves in monkey frontal and parietal areas encode recent reward history

Erfan Zabeh[1,2], Nicholas C. Foley[2], Joshua Jacobs [1,3,6] ✉ &
Jacqueline P. Gottlieb [2,4,5,6] ✉

Brain function depends on neural communication, but the mechanisms of this communication are not well understood. Recent studies suggest that one form of neural communication is through traveling waves (TWs)−patterns of neural oscillations that propagate within and between brain areas. We show that TWs are robust in microarray recordings in frontal and parietal cortex and encode recent reward history. Two adult male monkeys made saccades to obtain probabilistic rewards and were sensitive to the (statistically irrelevant) reward on the previous trial. TWs in frontal and parietal areas were stronger in trials that followed a prior reward versus a lack of reward and, in the frontal lobe, correlated with the monkeys' behavioral sensitivity to the prior reward. The findings suggest that neural communication mediated by TWs within the frontal and parietal lobes contribute to maintaining information about recent reward history and mediating the impact of this history on the monkeys' expectations.

Neural oscillations have long been proposed to regulate communication among neuronal ensembles within and across different brain structures[1,2]. Traditionally, neural oscillations have been interpreted as indicating so-called zero-lag synchrony, whereby large groups of neurons respond rhythmically with the same timing across cells. However, with the advent of multichannel recording technologies, mounting evidence shows that many oscillations are, in fact, traveling waves (TWs)−oscillatory patterns of activity that propagate across neural tissue at biologically plausible speeds consistent with axonal conduction velocities[3,4].

TWs have been found in multichannel recordings of local field potentials (LFPs) across multiple animal species, frequencies, brain states, and brain systems, suggesting that they are a widespread feature of neural activity. Importantly, TWs are found during sleep[5−7] and a variety of active behaviors−including in the hippocampus of rodents that are freely moving[8] or performing goal-directed navigation[9], visual regions of monkeys performing perceptual tasks[10,11] and the neocortex of humans performing visual[12−14] and memory tasks[15−17]. In some of these studies, specific TW properties (e.g., strength or direction)

correlate with accuracy and reaction times[10,17,18], suggesting that TWs are functionally significant in linking behavior with neural activity.

Given the near ubiquity of TWs, improving our understanding of their functional significance is of considerable interest. Because TWs activate ensembles of cells in succession, a key hypothesis is that they facilitate the integration of information across cells. The LFPs that form TWs reflect the average activity of the neurons underneath a local electrode[9,19]. Thus, a propagating TW indicates that there is a spatio-temporal pattern of neural activity−a wave of neuronal spiking−that is moving in a particular direction across broader populations of cells[20−22]. Supporting this view, studies proposed that TWs in topographically organized visual maps facilitate the integration of visual information across neurons encoding different retinotopic locations, providing support to the idea that TWs integrate information across space[10,23,24].

An open question, however, is whether TWs may also integrate information in time−e.g., by conveying information about recent events that influence the evaluation of future events. Temporal integration is particularly important for learning from recent rewards, a cornerstone of the mechanism for learning and decision-making.

[1]Department of Biomedical Engineering, Columbia University, New York, NY, USA. [2]Mortimer B. Zuckerman Mind Brain Behavior Institute, Columbia University, New York, NY, USA. [3]Department of Neurological Surgery, Columbia University, New York, NY, USA. [4]Department of Neuroscience, Columbia University, New York, NY, USA. [5]Kavli Institute for Brain Science, Columbia University, New York, NY, USA. [6]These authors contributed equally: Joshua Jacobs, Jacqueline P. Gottlieb. ✉e-mail: joshua.jacobs@columbia.edu; jg2141@columbia.edu

Animals, including humans and monkeys, are exquisitely sensitive to recent rewards and track these rewards in a so-called "model-free" fashion—even when they are statistically irrelevant and non-predictive of future outcomes[25–27]. Frontal and parietal areas contain neurons that encode recent rewards and are thought to be important for mediating the effects of reward history[27–31], but the mechanisms underlying this role are incompletely understood.

Here we examined this question by analyzing LFPs in microelectrode arrays implanted in the monkey lateral prefrontal cortex (LPFC) and parietal area 7A. Monkeys performed a task in which they made saccades to obtain probabilistic rewards, and although reward history was not predictive of the current trial reward, their behaviors were strongly sensitive to the prior trial reward. We show that, in both frontal and parietal areas, neural oscillations in the beta frequency band formed reliable TWs propagating in specific directions. Moreover, the strength of TWs was enhanced by receipt of a prior trial reward and, in the LPFC, reflected the influence of the reward on the monkeys' expectations, showing that TWs convey distinct information about recent reward history

## Results

### Monkeys are sensitive to irrelevant prior rewards

Two monkeys (*Macaca Mulatta*) performed a visually guided saccade task in which they obtained probabilistic rewards predicted by visual cues[28]. In each trial, the monkeys saw a visual cue specifying the trial's expected value (EV; the product of reward magnitude and probability) and, after maintaining fixation for an additional delay period, made a saccade to a target to obtain the reward (Fig. 1A). The cue and target stimuli each appeared at two possible locations to the right and left of fixation. Their locations were independently randomized so that the cue signaled reward expectations but not the saccade motor plan (see "Methods" for details).

We recorded the monkeys' anticipatory licking as an index of their reward expectations. After the presentation of the reward cue, licking rates increased with the EV signaled by the cues, confirming that the monkeys were familiar with and attentive to the cues (Fig. 1B, right). Surprisingly, although the reward on the current trial could not be predicted from that on a previous trial, licking was also highly sensitive to the prior-trial reward. During an early period before the onset of the current reward cue, licking was more vigorous if the prior trial ended with a reward (PR) versus a lack of reward (PNR; Fig. 1B, left). A significant influence of the previous outcome persisted after cue onset when it co-existed with the effect of EV (Fig. 1B, right). The prior trial effect on licking did not merely reflect consumption of the reward because licking ceased during the inter-trial interval and resumed at the start of the next trial ("Methods"; Foley et al., 2020). Licking rates significantly correlated with rewards on the previous trial (Mj: $r = 0.342$, $p < 0.001$; Mc: $r = 0.332$, $p < 0.001$) but not with those further removed in the past (2 trials back, monkey Mj: $r = 0.001$ $p = 0.46$; monkey Mc: $r = 0.003$ $p = 0.43$; 3 trials back, all $p$'s > 0.8). The robust effect of the prior trial reward is consistent with findings that humans and monkeys are highly sensitive to recent reward history even when this history is irrelevant (unpredictive) of future rewards[26,29,32].

### Oscillatory activity in the LPFC and PPC shows TWs

To understand the neural basis of the prior trial effect, we recorded activity using electrode grids implanted in the dorsolateral prefrontal cortex (LPFC) and posterior parietal cortex (PPC) in each monkey (Fig. 2A)[33,34]. Analysis of LFPs showed that many electrodes showed prominent oscillatory activity in the beta-band (-15 Hz). In addition, across electrodes, the oscillations had phase shifts with a consistent spatial gradient, suggesting the presence of TWs, or plane waves that propagated in consistent directions. Figure 2B shows an example of this phenomenon in one trial from the PPC of monkey Mc. The LFP signals showed an oscillation waveform that appeared similar on

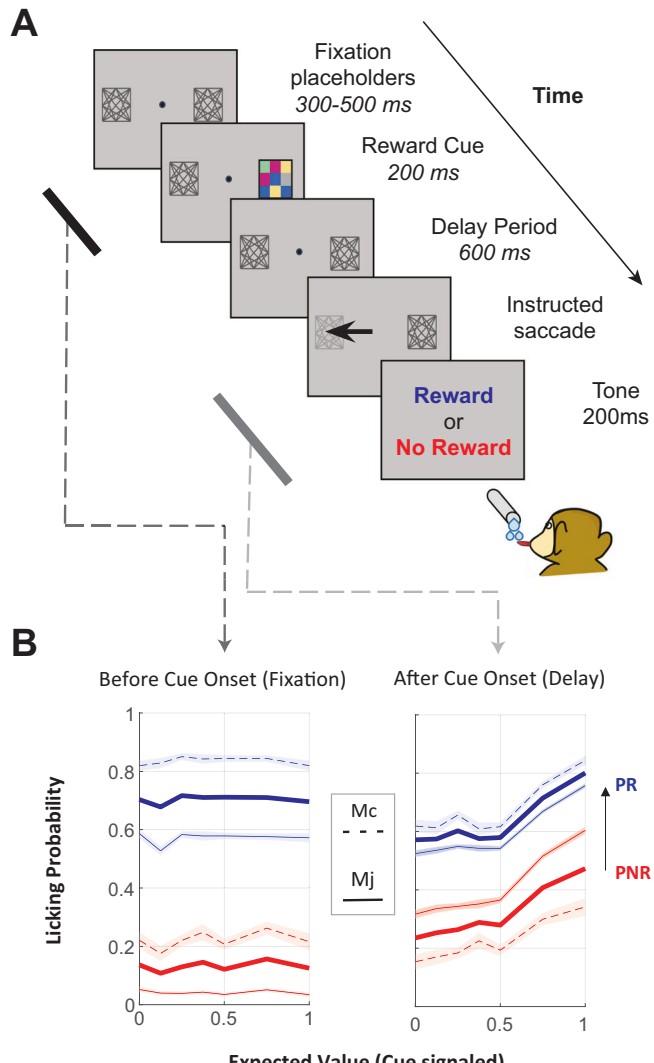

**Fig. 1 | Task and behavior. A** Timeline of trial events. On each trial after achieving fixation, the monkeys saw a reward cue colored checkerboard) indicating the trial's EV. After a 600-ms delay period, a saccade target appeared, and after making the required saccade, the monkeys received the outcome (reward or lack of reward) according to the cued probability. Two gray laceholders were continuously present on the screen, marking the possible locations of the cue and target (which were randomized independently across trials). **B** Probability of anticipatory licking as a function of EV and prior trial reward. The traces show the king probability as a function of EV during two task epochs: an early pre-cue epoch (left, the 200 ms epoch ending at reward cue inset) and a late post-cue epoch preceding reward delivery (right: 400 ms epoch prior to reward onset). Blue traces indicate trials that followed a prior reward (PR), and red indicates trials that followed a prior no reward (PNR). The thin traces show the mean and EM across all correct trials for each monkey (Mj PR, $N = 7685$; Mj PNR, $N = 4341$; Mc PR, $N = 4023$; Mcj PNR, $N = 2104$) and thick traces show the average across monkeys. The effect of the prior reward was significant in each monkey and epoch (linear regression, all $p < 0.001$).

neighboring electrodes (Fig. 2B, top two rows) but showed a systematic phase shift across adjacent electrodes when measured quantitatively (Fig. 2B, bottom row, electrodes 1–5). The phase shift had a consistent orientation across the array, suggesting that the oscillation propagated in an anterior-to-posterior direction across the electrode grid (Fig. 2C).

To quantitatively measure TWs, we used circular–linear statistics[35] to extract the phase gradient directionality (PGD) index[17] (see "Methods"; Supplemental Movie S1). The index measures TW strength—the consistency with which oscillations propagate in a specific direction—and we thus refer to it interchangeably as PGD or wave strength. A PGD

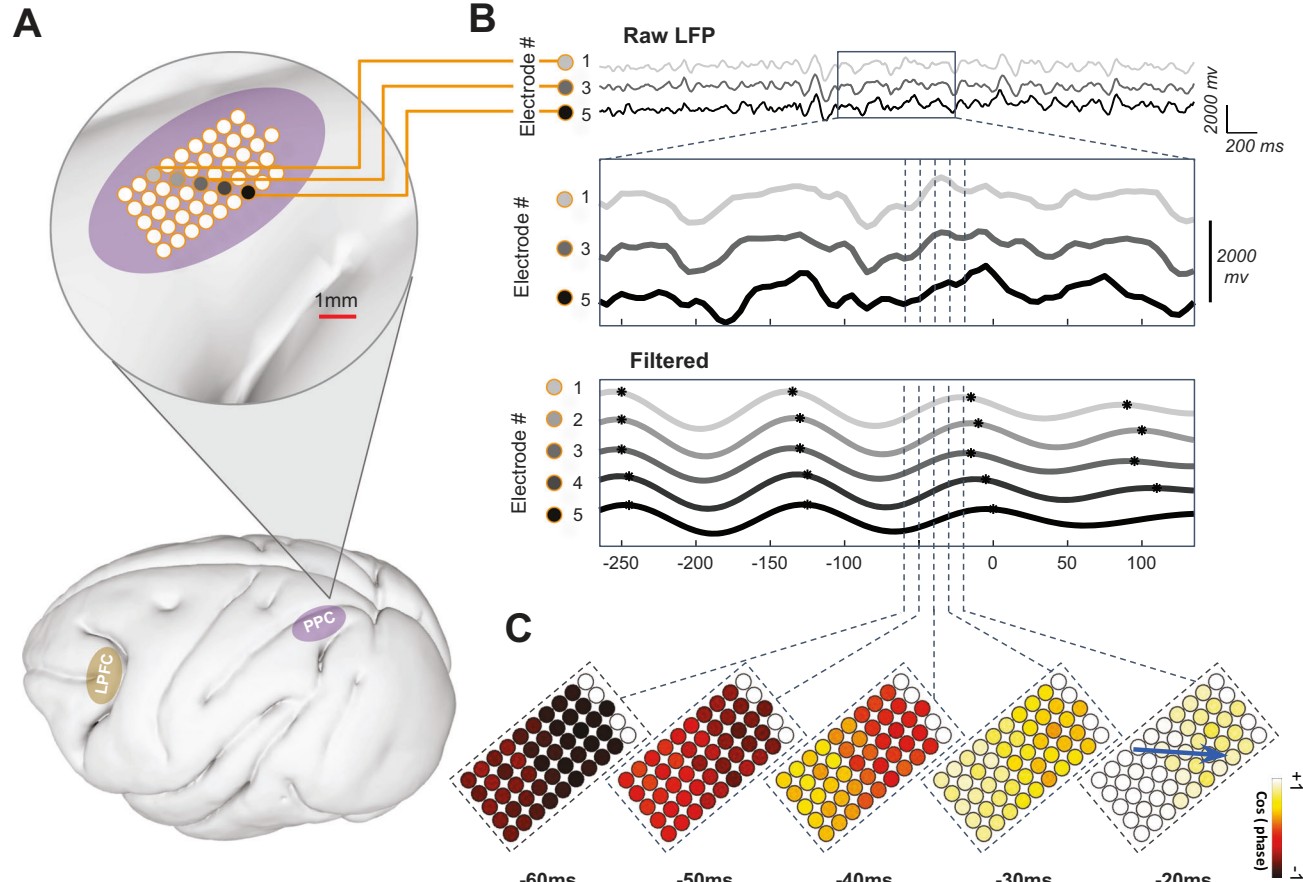

**Fig. 2 | Spatial organization of oscillations and traveling waves (TW) in LPFC and PPC. A** Locations of implanted Utah microarrays. The bottom panel shows the locations of the PPC (purple) and LPFC (yellow) grids, as estimated from stereotactic coordinates and intra-operative photographs and projected on a canonical 3D lateral view of the cortex from the diffusion tensor MRI atlas of the *Rhesus macaque* brain[33,34]. The top panel shows a closeup of the PPC array from Monkey Mc, highlighting the electrodes analyzed in (**B**). **B** A TW in a representative trial. The top panel shows the unfiltered continuous LFP signals from every other electrode in the group highlighted in (**A**). The middle panel is an expanded view of the same LFP traces, focusing on a smaller time window aligned on cue onset (0 ms). The bottom panel shows the signals at all five electrodes filtered at 14 Hz (Bandwidth 1.5 Hz). The peak phases of these oscillations (black dots) occur at successively later times for electrodes 1–5, thus demonstrating a TW. **C** Visualization of the TW across the recording array. The color shows several snapshots of the relative (cosine) phase of the 14-Hz LFPs across the array at 10-ms intervals. The blue arrow indicates the TW direction at −20 ms.

with a value near 1 indicates high wave strength, i.e., a TW with a highly coherent propagation pattern across the electrode grid. In contrast, a PGD of 0 indicates the absence of a TW—e.g., a situation in which there is high LFP power, but the phase shifts show random variations that are not spatially organized across the electrode grid. Similarly, we calculated the speed and direction of the TW ("Methods").

To analyze the properties of the TWs, we calculated the mean PGD at each frequency between 2 and 50 Hz at time points from 1.2 s before to 2 s after cue onset. This showed that TWs were robust throughout the pre-cue, cue, and delay epochs and were most prominent in the alpha-beta frequency band in both areas (10–30 Hz; Fig. 3A), with the frequency showing the largest PGD values being positively correlated with the frequency showing the highest LFP power (Fig. 3C; see "Discussion").

The directions of the TW propagation were not random but were oriented along an axis approximately perpendicular to the nearby sulci —i.e., the principal sulcus in the LPFC and the intraparietal sulcus for the PPC (Fig. 3E). TWs were equally likely to occur in both directions along these axes, resulting in a bimodal distribution, with peaks in anterior-dorsal and posterior-ventral directions (non-uniformity of circular distributions: $p < 0.001$ for each array, Omnibus circular test). For all propagation directions, the speed of TW propagation was 0.1–0.6 meters/second, consistent with conduction velocities in unmyelinated axons[36,37] (Fig. 3D).

Although TWs were pronounced in both the frontal and parietal areas, their properties significantly differed between the two areas. The frequency showing the strongest TWs was significantly higher in the LPFC relative to the PPC (Mc: 19.1 Hz vs. 15.6 Hz; Mj:13.6 Hz vs. 11.5 Hz; all $p$'s $< 0.001$, rank-sum test; Fig. 3A–C). TW speed was higher in the LPFC relative to the PPC (Wilcoxon rank-sum test: $p < 0.001$). Finally, the directional distributions were significantly broader in the LPFC relative to the PPC (the mean and standard error (SEM) of the angular distance from the main axis of the distributions was, in Mc, PPC: $26.93 \pm 0.03°$ vs. LPFC: $37.88 \pm 0.04°$, $p < 0.01$; and in Mj, PPC: $26.64 \pm 0.03°$ vs. LPFC: $43.58 \pm 0.04°$, $p < 0.01$, circular Kuiper test). These differences suggest that the frontal and parietal TWs are more likely to reflect local activity within each lobe rather than a single extended TW that propagates across the fronto-parietal network (see "Discussion").

**Pre-cue TW strength correlates with PRs**

Analyses of the relation between TW properties and behavior showed that TW strength was specifically modulated by the reward the monkeys received in the previous trial. Figure 4A illustrates this result in two representative trials from the PPC of monkey Mj. The trial in the top panel occurred immediately after a trial in which the monkey had received a reward; the trial in the bottom panel occurred immediately after a trial in which the monkey received no reward. In the PR trial (Fig. 4A, top panel), during the initial fixation period before the reward

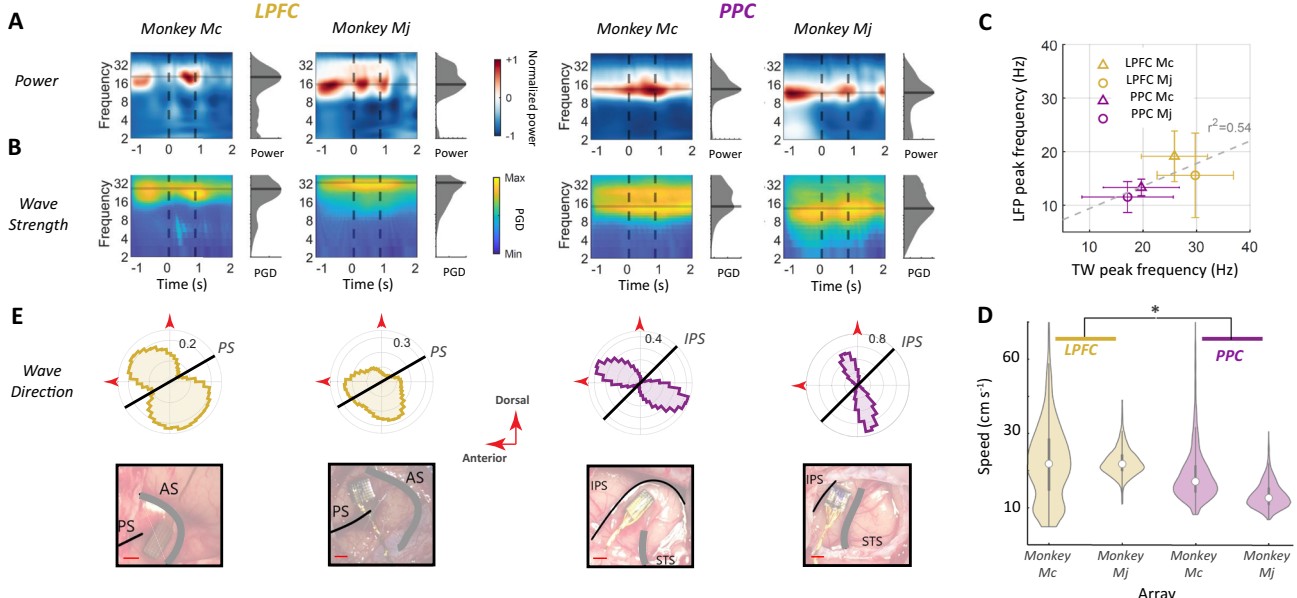

**Fig. 3 | TW properties. A** LFP power spectrograms. Wavelet spectrograms were computed independently for each electrode and trial and averaged across each array (left: LPFC; right: PPC). Power is normalized based on the maximum power pixel value in each array. The black dashed lines show, respectively, cue onset and target onset. The gray plot shows the average LFP power as a function of frequency, and the shaded horizontal lines indicate the peak frequency. Peak LFP power was in the beta-band in all arrays and task epochs. **B** Spectrogram of TW wave strength. Wave strength (PGD) was calculated on each trial at centering frequencies from 2 to 50 Hz. The gray marginal histograms show the time-averaged wave strength as a function of frequency. The color codes are calculated independently for each array. **C** Correlation between the peak frequencies of TWs and LFP power (regression, $r^2 = 0.54$, $p < 0.01$, two-tailed test). Each point shows the mean and standard deviation of the peak frequency for LFP power and wave strength in individual trials for each array. **D** TWs speeds. The violin plots show the distributions of TW speeds

(averaged over the interval of −1.2 s to 2 s relative to cue onset at the peak frequency (LPFC: Mc 25.7 Hz, Mj 29.7 Hz; PPC: Mc 19.7 Hz, Mj 17.1 Hz). The plots extend to the full range of the data; the white dots show medians, the thin whiskers show 99% coverage, and the thick whiskers show the standard deviations. Star indicates faster velocities in the LPFC (rank-sum test: *$p < 0.001$). **E** Distribution of TWs propagation directions. The bottom row shows intraoperative photographs showing the locations of the arrays relative to sulcal landmarks: the principal sulcus (PS) and arcuate sulcus (AS) for the LPFC, and the intraparietal sulcus (IPS) and superior temporal sulcus (STS) for PPC. In the top row, the circular histograms show the distribution of TW directions across all recording trials (measured as in **D**). The red arrowheads show the dorsal and anterior directions, and the black lines denote the axes of the PS and IPS. Panels **A**–**E** are based on $N = 4368$ trials (LPFC Mc), $N = 8105$ (LPFC Mj), $N = 4567$ (PPC Mc), and $N = 3605$ (PPC Mj).

---

cue was presented, the propagation gradients at individual electrodes were organized in consistent directions, resulting in a TW propagating towards 180° (PGD = 0.42). In contrast, in the PNR trial (bottom panel), TW propagation directions were inconsistently organized across electrodes, resulting in a low TW strength (PGD = 0.12).

Confirming these individual-trial examples, TWs in pre-cue periods were stronger on PR than PNR trials in each area and monkey (Fig. 4B, C). Two-tailed Wilcoxon rank-sum tests revealed that TW strength was significantly higher for PR than PNR trials in all cases (LPFC: Mc, $p < 0.01$, Mj, $p < 0.05$, PPC: Mc, $p < 0.01$, Mj, $p < 0.01$), with effect-sizes measured in $d'$ ranging between 0.12 and 0.31 (LPFC, Mc, $d'$ 0.26; Mj, 0.12; Mc, 0.31; Mj, 0.24). To examine the time and frequency range of this effect, we used a generalized linear model (GLM) to fit PGD strength as a function of the prior-trial reward across frequencies and time points (Fig. 4D). The coefficients indicating the prior trial effect were significant in a frequency range near the dominant frequency of the TW and during a time window limited to the fixation epoch preceding cue onset (i.e., −1000 to 0 ms in Fig. 4D, E).

To rule out the possibility that these apparent links between TW strength and reward were due to confounds, we next fit the average PGD strength in this time-frequency range with an expanded GLM that not only modeled prior trial reward as before but also included additional regressors for pre-cue licking rate and prior-trial EV. The prior trial regressor continued to produce significant positive coefficients in all arrays (coefficients: 0.09, 0.16, 0.23, and 0.26, respectively, for the LPFC of Mc and MJ and the PPC of MC and MJ, all $p$'s < 0.001; *Methods*) and accounted for the vast majority of the total variance explained by

the GLM (variance explained by prior trial regressor: LPFC: Mc 79.3%, Mj: 90.4%; PPC: Mc 99.7%, Mj 94.9%). In contrast, the coefficients for the other covariates explained much less variance than prior trial reward (variance explained, pre-cue licking: 19.9%, 7.7%, 0.1%, and 4.7%; prior-trial EV: 0.7%, 1.8%, 0.2%, and 0.3%) and were themselves not significantly above chance (prior trial EV, $p > 0.35$ for all arrays; pre-cue licking, $p > 0.3$ in LPFC of monkey Mj and PPC of monkey Mc). Thus, TW strength significantly encoded the outcome of the previous trial, and there was no evidence that this effect was driven by confounds.

Analyses of other task epochs showed that PGD did not change upon reward delivery (see black line in Fig. 4E and Fig. S4A), ruling out that the prior-trial effects were merely passive continuations of a response to the previous outcome. Moreover, during the post cue (visual and delay) epochs, there were no significant modulations related to the cue or target locations, ruling out visuo-spatial confounds (Fig. S4B). Finally, PGD was not sensitive to the current trial EV (Fig. 4E; Fig. S4A), indicating that they conveyed specific information about the recent reward history independently of other reward feature.

## TW strength predicted PRs independently of other physiological markers

As shown in Fig. 3, TWs showed not only variable strength but also variable speed and direction, and their peak frequencies correlated with those of LFP oscillations. This raises the question of whether the association between PGD and PR remained significant after controlling for additional properties of TWs and LFP oscillations. To examine this

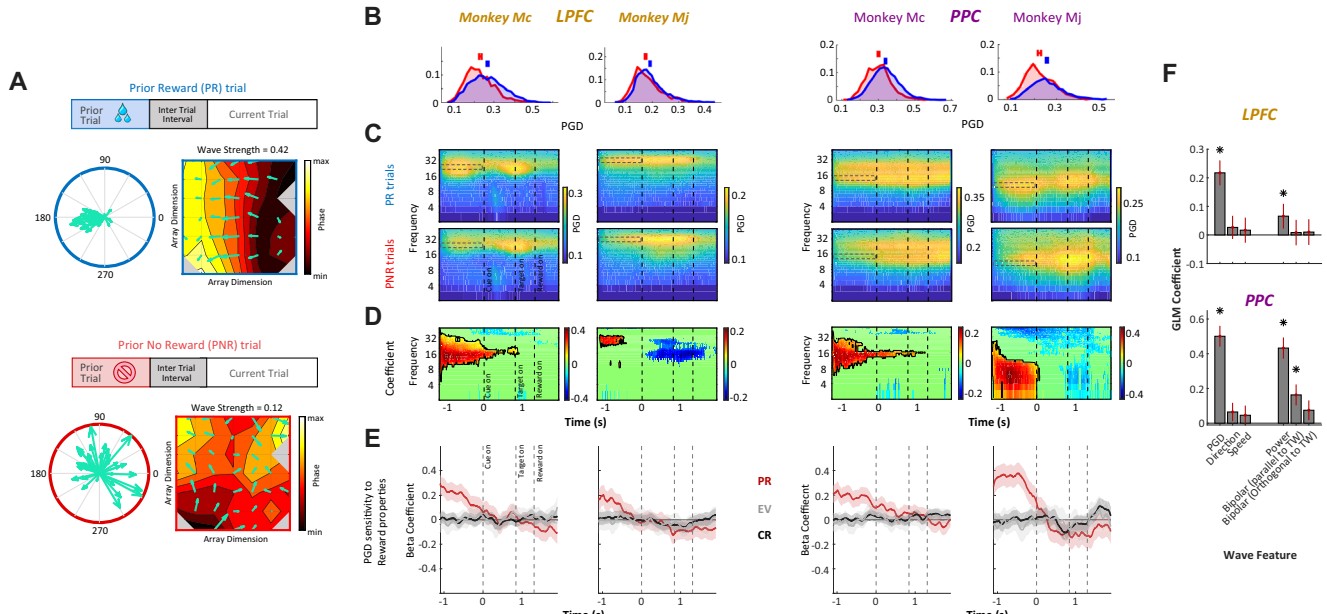

**Fig. 4 | TW strength depends on the prior reward. A** Two representative trials from Mj PPC, which followed a prior reward (PR, top) and a prior no-reward (PNR, bottom). The color maps show the phase gradient across the array during the 100ms preceding cue onset. Arrows show the phase gradient vector at each electrode location, and lines show the 8 Hz phase contours. The left polar plots are circular histograms of the phase gradient vectors; TW strength increases with the directional consistency gradient vectors. **B** Distributions of TW strength in the pre-cue epoch (dashed box in panel **C**) for PR (blue) and PNR (red) trials. The horizontal bars are centered at the mean, and their width is equal to 2 SEM. **C** PGD indices versus time and frequency. All conventions as in Fig. 3A. **D** GLM coefficients for the prior trial effect from the maps in (**B**). Coefficients with $p > 0.05$ (F-test) are depicted as 0. The black contours show the regions of interest from which we derived the distributions in (**A**), defined as clusters of adjacent pixels with significant positive coefficients. The earliest time measured (−1200 ms) is the median

fixation duration; the clusters extend to this point because of trials with longer fixation durations. **E** GLM coefficients indicate the effect on the PGD of the prior trial outcome (PR, red), expected value (EV, gray), and current reward (CR, black) (see Fig. S4). Each trace shows the coefficients (solid) and their 95% confidence intervals (shaded) at the central frequency of the region of interest in (**D**) (LPFC: Mc 22 Hz, Mj 32 Hz; PPC: Mc 12 Hz, Mj 10 Hz). **F** Relation between PGD and reward history. GLM coefficients predicting the prior trial outcome based on wave strength, propagation direction, propagation speed, average LFP power, and bipolar LFP for pairs of adjacent electrodes lying parallel and orthogonal to the TW direction. Red error bars represent two standard errors of the regression coefficients, and stars represent significant coefficients (95% confidence interval does not include 0, equivalent to $p < 0.05$, two-tailed). Panels **A**–**F** are based on the following N's of independent trials: LPFC Mc: PR 2852, NPR 1516; LPFC Mj: PR 5165, NPR 2940; PPC Mc: PR 2987, NPR 1580; PPC Mj: PR 2681, NPR 924.

question, we used a multivariate approach to simultaneously model prior trial reward as a function of several TW features−TW strength (PGD), speed, and direction−as well as the power of LFP oscillations. In addition, we included alternative measures of TW propagation that did not use circular statistics−the bipolar-LFP, defined as the instantaneous voltage differences between two adjacent electrodes that were oriented parallel or perpendicular to the main direction of TW propagation ("Methods").

In this multivariate model, all the TW and LFP features competed to predict PR, and the regressors were standardized. Thus, the significance and magnitude of the fitted coefficients show the unique variance explained by each measure. As shown in Fig. 4F, the largest coefficients corresponded to PGD and were significant in all grids, whereas the coefficients for TW direction and speed and most of the bipolar LFP measures did not significantly differ from zero. Several LFP measures did produce significant coefficients, including the parallel bipolar LFP in the PPC and LFP power in each area, but these coefficients were significantly smaller than the PGD coefficients (all $p$'s < 0.05, rank-sum tests). To quantitatively measure these differences, we computed the variance explained by each factor relative to the model as a whole ("Methods"). PGD strength was the strongest predictor of prior trial reward, explaining 75.8% of all variance explained in the LPFC and 39.8% in the PPC. LFP power accounted for a much smaller amount of variance (respectively, 6.8% and 27.2%), and all the other metrics accounted for less than 4%. Thus, TW strength explained a substantial unique portion of the variance in explaining PR that was not explained by other features of TWs and LFP oscillations.

## TWs in the LPFC encode the prior trial effect on the monkeys' expectations

Given the specific association between PGD and the prior trial reward, we asked if this modulation predicted the animals' behavioral sensitivity to the PR. To examine this question, we focused on the monkeys' licking response during the pre-cue epoch as an index of their sensitivity to the PR and used a receiver operating characteristic (ROC) analysis to identify trials in which licking rates were consistent versus inconsistent with the PR (*Methods*). This analysis identified trials in which the monkeys showed licking consistent with the prior trial reward (high licking on PR trials and low licking on PNR trials), suggesting that they correctly *Retrieved* the PR (Fig. 5A, pale green) and trials in which they showed inconsistent licking (low licking on PR trials and high licking on PNR trials), suggesting they *Neglected* the PR (Fig. 5A, gray/dark green).

We reasoned that a relation between PGD and behavior would manifest as a statistical interaction, whereby the PR modulation in the PGD would be larger when licking was consistent versus inconsistent with the PR. This effect was visible in the fit from the GLM model in the LPFC (Fig. 5B), as the coefficients measuring the PR effect were significantly larger on Retrieved vs. Neglected trials (Mj and Mc: $p < 0.01$). To confirm the interaction effect directly, we fit an alternative model of PGD that included regressors for PR, behavior (Retrieved/Neglected), and their multiplicative interaction. In the LPFC, the PR and interaction terms each produced significant coefficients and accounted for more than 30% of the explained variance in all cases (PR: Mc: 50%, $p = 0.007$; Mj: 42%, $p = 0.09$; interaction: Mc: 33%, $p = 0.03$; Mj: 49%, $p = 0.06$; Full model fit: Mc $R^2 = 0.080$; Mj $R^2 = 0.017$, all F-test $p$'s $< 10^{-4}$). In contrast,

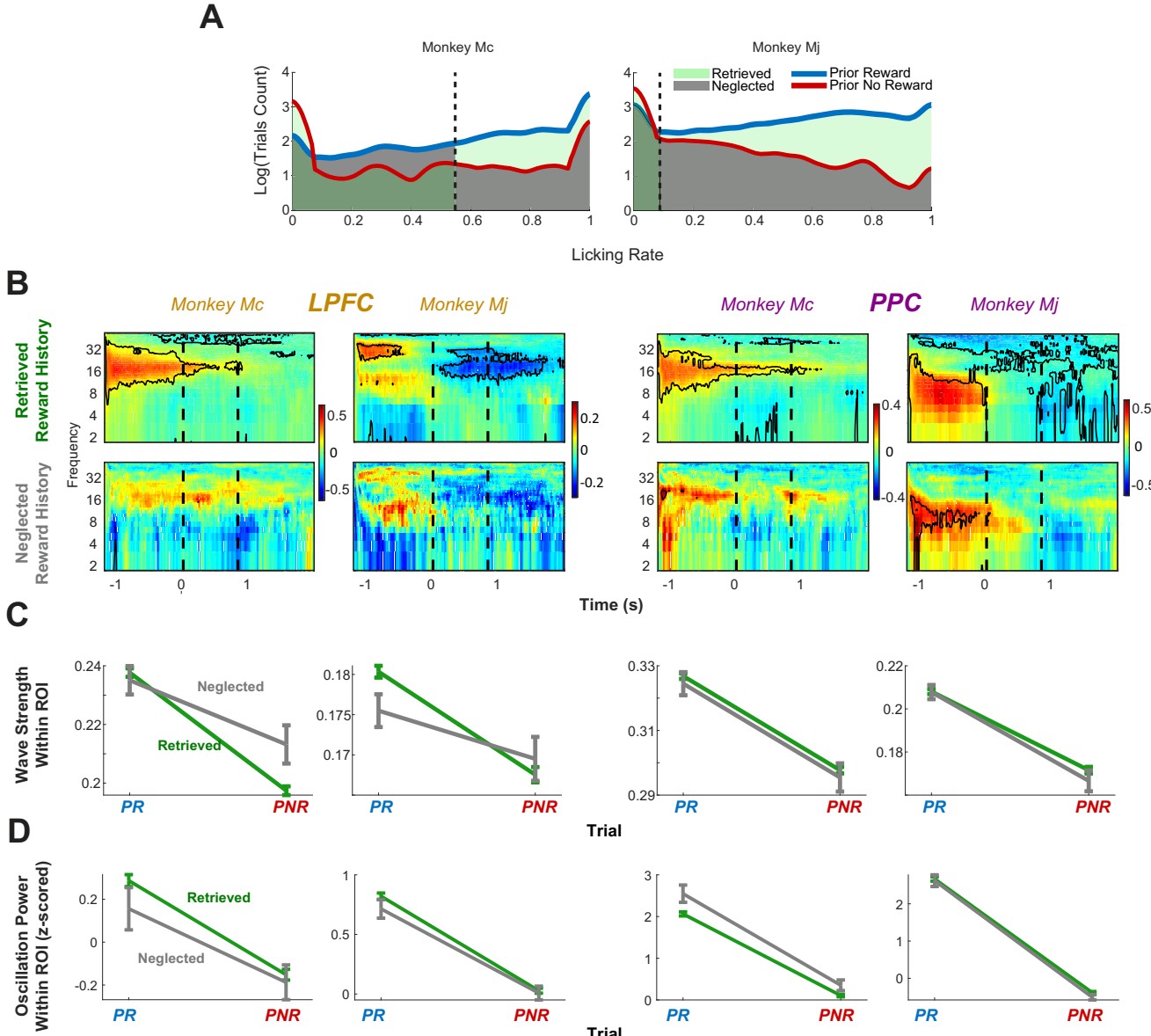

**Fig. 5 | TW strength encoded behavioral sensitivity to the prior reward.**
**A** Relation between licking rate and prior trial reward. The blue and red traces show the distributions of licking rates during the pre-cue epoch (500 ms to 0 ms before cue onset) for, respectively, PR and PNR trials. The dashed vertical line is the decision criterion that afforded maximal discrimination of the prior reward based on ROC analysis ("Methods"). Pale green shading shows the trials in which the monkey licked consistently with the prior reward suggesting that they retrieved this reward, and dark gray shows trials in which they licked inconsistently with the prior reward, suggesting that they neglected the prior reward. **B** Relation between prior reward and PGD as a function of the monkey's behavior. Spectrograms of GLM coefficients for the effect of the prior reward for retrieved and neglected trials; same format as Fig. 4D. **C** Summary plot showing the relation between TW strength in the LPFC and behavior. PGD as a function of prior trial outcome (abscissa) and licking (green vs. gray traces). Data points represent the mean and SEM of averaged PGD values in the pre-cue time–frequency interval of interest over all corresponding trials. Sensitivity to behavior is seen as an interaction, whereby the slopes are higher for retrieved versus neglected trials, which is significant for the LPFC but not PPC in both monkeys. **D** Relation between LFP power and behavior. z-scored LFP power (mean and SEM) in the pre-cue region of interest in the same format as (**C**). Analyses in panels **A**–**D** are based on trials with valid LFP and licking measurements, with the following N: LPFC Mc [2294 PR Retrieved, 1189 NPR Retrieved, 168 PR Neglected, 96 NPR Neglected]; LPFC Mj [4079 PR Retrieved, 2257 NPR Retrieved, 497 PR Neglected, 263 NPR Neglected]; PPC Mc [2788 PR Retrieved, 1148 NPR Retrieved, 200 PR Neglected, 95 NPR Neglected]; PPC Mj [1910 PR Retrieved, 650 NPR Retrieved, 241 PR Neglected, 73 NPR.

there was no significant effect of the Retrieved/Neglected factor alone (variance explained, Mc: 17%, $p = 0.1$; Mj: 9%; $p = 0.4$), which is consistent with our interpretation that the behavior was evident in the LPFC PGD by modulating the strength of the prior trial effect. As additional confirmation, we conducted a session-level analysis in which we split the sessions into two equal groups based on a median split between their correlation between licking and PR. The PR coefficients for PGD were higher in sessions with higher behavioral sensitivity to PR, consistent with the trial-level results (Fig. S5).

These behavioral modulations were specific to the PGD of TWs in LPFC. Similar effects were not present in the PPC, where the behavior and interaction terms were not significant ($p$'s > 0.4; <3% variance explained). An ANOVA analysis with factors for area (LPFC/PPC), PR, and behavior (Retrieved, Neglected) produced significant 3-way interactions in each monkey between PR × area × behavior (Mj < 0.05, Mc < 0.01), confirming that the effect was stronger in the LPFC. Moreover, LFP power showed no significant behavioral modulations in any array (Fig. 5D; GLM p's for interaction effect: all >0.3). Thus, the

relation between PR and the animal's behavioral sensitivity was only reflected by the strength of TWs in the LPFC and not by TW's strength in the PPC nor the power of LFPs in either region.

## Discussion

We show that TWs of beta-band oscillations are present in the monkey frontal and parietal cortex and convey information about recent rewards. These results are consistent with previous evidence implicating beta-band oscillations in cognitive control, working memory, and reward computations[38–43] but suggest that at least some of the beta-band oscillations in frontal and parietal areas reflect signals that propagate in specific directions as planar TWs[44]. Moreover, the correlations between TW strength and PRs show that, in addition to their associations with motor function, navigation, and memory[4,8–10,17,18,22,45–48], TWs convey information about reward processing.

Behavioral sensitivity to statistically irrelevant PRs like that shown by our monkeys has been well-documented in several species and may contribute to reward learning in some contexts while producing decision biases in others[26,32]. While the neural underpinnings of this sensitivity are not fully understood, our results provide several specific clues about the possible contributions of TWs. First, we show that TWs specifically signaled the prior trial reward or lack of reward rather than the expected value of the reward on the current or previous trial. Second, TWs encoded PR information de novo at the start of a trial rather than responding to reward delivery and maintaining this response through the following trial. Third, TWs encoded the PRs transiently before the presentation of the current trial's reward cue. Finally, TW strength in the frontal lobe correlated with the impact of the PR on the monkeys' future reward expectations as indexed by anticipatory licking behavior. Together, these properties suggest that TWs in the frontal and parietal lobes contribute to the active retrieval of reward history information at the start of a trial and, in the frontal lobe, also signal the use of this information to adjust future expectations. Interestingly, the effect of behavior seemed to differ by a monkey, since in monkey Mj, the primary effect of successful retrieval was the enhancement of TW after a PR, while in monkey Mc it was the suppression of TW strength after a prior no reward (Fig. 5C). We hypothesize that these differences may reflect the monkeys' behavioral strategies (perhaps their attention to different task attributes), an idea that can be examined with future behavioral tests.

Recent results from our laboratory suggest a potential mechanism by which these TWs influence behavior. We found that, in the same behavioral task, individual cells in these areas responded to the delivery of an outcome (a "current reward") in ways that depended on the PR[28]. The highest firing rates were evoked when the current trial failed to deliver a reward but followed a prior trial that did produce a reward−i.e. when there was a *decline* in reward rate across consecutive trials[28]. Based on the role of the frontoparietal network in executive monitoring, these findings suggest that its neurons signal downstream areas to increase cognitive effort upon detecting a decline in reward rates[49–51]. We can speculate, therefore, that these "requests for adjustment" are carried by spiking responses, which represent the output of an area; in contrast, TWs may be part of the mechanism for producing these responses. TW strength, which transiently encoded PRs at the start of a trial, may produce sustained changes in neuronal excitability that last until the end of the trial and facilitate the comparison of the current and PRs (consistent with Foley et al. 2020)[28]. An important question for future research is thus whether and how TWs interact with spiking activity to set the context for evaluating a current reward−more specifically, facilitate comparisons between sequential outcomes for the purpose of monitoring reward rates and triggering adjustments in behavior.

Previous studies have shown that individual cycles of neural oscillations and TW are coupled to the spiking of individual neurons[19,52–56]. Thus, a spatially propagating TW produces neuronal

spiking and/or subthreshold activity that moves across the cortex in a consistent direction[21,22], providing a mechanism by which multiple features of a TW, including their strength and direction, can be physiologically and behaviorally significant. The literature provides mixed results about the specific aspects of TWs that are consequential, with some studies showing that behavior is more strongly correlated with TW propagation direction[12,57–60] and others showing stronger behavioral correlations with TW strength[10,21,47,61,62].

Our results combine elements of both views. We show that PRs enhanced only the strength of the TWs without changing their propagation direction, suggesting that TW strength specifically encoded the previous outcome. However, rather than being uniformly distributed, TWs propagated primarily in two opposite directions along a single spatial axis, suggesting an additional role for propagation direction. This pattern may reflect the joint action of two mechanisms. Because TWs are associated with spatial gradients in synaptic weights[11], the non-uniform directional distributions may reflect preferential anatomical connectivity along axes that are orthogonal rather than parallel to a sulcus[47]. The modulations of TW strength, on the other hand, may reflect the upregulation of general network processes, like the strengthening of local phase coupling[3] or an overall increase in inhibitory activity[11], which, when combined with the connectivity gradients, may preferentially strengthen TWs along the connectivity axes.

Decoding cognitive variables from neural oscillations can be challenging[63–66] because multiple oscillation features, such as power, phase, speed, and direction, can appear intertwined. We show that the peak frequencies for LFP power and TW strengths were correlated. Note that although Fig. 3C suggests that the frequencies of power peaks were lower than the frequencies of peak PGD, this may have been a measurement issue caused by the $1/f$ tilt in the LFP power, which causes power peaks to be extracted at lower frequencies[67,68], without affecting the frequencies extracted for phase or TWs. Thus, these results are largely consistent with the notion that there is a single underlying oscillation whose power and TW strength vary independently (as explained in Fig. 1), although it remains theoretically possible that there are separate oscillators underlying the power and TW strength effects. We used multivariate modeling to statistically distinguish the contributions of different TW features to encoding PRs and confirmed that TW strength conveys information that is independent of TW speed and direction, as noted above, and, importantly, is also independent of LFP power. The PGD of beta-band TWs remained the strongest predictor of reward history even after controlling for LFP power and, in the prefrontal cortex, uniquely predicted the monkeys' behavioral sensitivity. This strongly supports our conclusion that measuring TWs provides additional useful information that supplements established analysis methods for oscillations at individual electrodes.

We further examined the issue of how best to measure TWs by comparing our primary methods based on circular statistics versus a bipolar LFP measure that is often used for measuring cortical gradients[54,69]. Because TWs propagate in particular directions across the cortex, a bipolar voltage differential across a pair of adjacent electrodes in the direction of propagation can measure a TW. We considered two forms of bipolar LFPs, with contacts referenced parallel and perpendicular to the main direction of TW propagation. However, the PGD proved to be a more reliable marker than both of the bipolar measures in all areas. The bipolar-measured signal was significant only in the PPC, and even there, it showed a much smaller coefficient relative to that of the PGD. We hypothesize that the PGD provided a more accurate measure of the TW's properties because it captured a more general pattern of LFP gradients across the entire grid rather than just leveraging a single bipolar electrode pair.

It is important to note that because TWs can propagate over large cortical distances, the TWs we measured with our microarrays may reflect a partial view, through narrow ~10 mm² apertures, into larger and more complex patterns[3,39]. In our data, the peak frequencies of

TWs differed between the frontal and parietal areas. Because of this difference, it is likely that the TWs we measured were restricted to each area rather than being part of a larger TW pattern that propagates across the fronto-parietal network. Nonetheless, an important direction for future research is to measure the spatial properties of oscillations on *both* small and large scales to understand how the local, micro-scale TWs we describe relate to larger mesoscale circuits.

In sum, our findings show that TWs reflect distinct aspects of reward computations relative to the information represented by the power of local LFP oscillations. This emphasizes the importance of understanding the spatio-temporal organization of LFP oscillations across multichannel recording arrays beyond conducting univariate measurements from individual electrodes[4]. The spatial organization of oscillations and TWs may reflect fundamental aspects of neural computation. For example, some recent studies suggested that TWs are specifically important for neuronal computations involving spatially distributed neural assemblies (e.g.,[70,71]), and our findings suggest that they may also be fundamentally informative for explaining how the brain supports types of neural computations involving events that are distributed in time. Going forward, leveraging their distinctive and widespread correlations with various aspects of behavior, TWs, and spatial patterns of oscillations may be important practically for brain-computer interfaces. Our demonstration that TW strength conveys unique behavioral information that is not present in LFP power suggests that a wide range of brain-computer interface challenges may benefit from measuring the spatial propagation of oscillatory activity for improved decoding of variables relevant to behavior and cognition.

## Methods

### General methods

Data were collected from two adult male rhesus monkeys (Macaca mulatta; 9–12 kg). The monkeys were motivated to perform the task by fluid restriction, and their weights and health were monitored daily. All methods were approved by the Animal Care and Use Committees of Columbia University and New York State Psychiatric Institute as complying with the guidelines within the Public Health Service Guide for the Care and Use of Laboratory Animals. Visual stimuli were presented on an MS3400V XGA high-definition monitor (CTX International, INC., City of Industry, CA; 62.5 by 46.5 cm viewing area). Eye position was recorded using an eye-tracking system (Arrington Research, Scottsdale, AZ). Licking was recorded at 1 kHz using an in-house device that transmitted a laser beam between the tip of the juice tube and the monkey's snout and generated a 5 V pulse upon detecting interruptions of the beam when the monkey extended his tongue to obtain water.

### Task

A trial started with the presentation of two square placeholders (1° width) located along the horizontal meridian at 8° eccentricity to the right and left of a central fixation point (white square, 0.2° diameter). After the monkey maintained gaze on the fixation point for 1300–1500 ms (fixation window, 1.5–2° square), a randomly selected placeholder was replaced for 300 ms by a reward cue—a checkerboard pattern indicating the trial's reward contingencies. After a 600 ms delay period, the fixation point disappeared simultaneously with an increase in luminance of one of the placeholders (the target), whose location was randomized independently from that of the cue. If the monkey made a saccade to the target with a reaction time (RT) of 100–700 ms and maintained fixation within a 2.0–3.5° window for 350 ms, he received a water reward with the magnitude and probability that had been indicated by the cue. An auditory tone (200 ms, 500 Hz) signaled the end of the post-saccadic hold period on all trials, providing a temporal marker for the onset of the outcome/ITI period whether a reward was received or omitted. Rewards, when delivered,

were linearly scaled between 0.28 and 1.12 mL. The ITI—from tone onset to the onset of the fixation point on the following trial lasted for 1200–1600 ms. Error trials (resulting from fixation breaks, premature, late, or wrong-direction saccades) were immediately repeated until correctly completed, precluding the monkeys from aborting trials in which they anticipated lower rewards.

Monkeys were extensively familiarized with the task and all the cues before recordings began. The full set was comprised of 20 distinct cues, including cues that indicated rewards with maximal size at 0.25, 0.5, or 0.75 probability, rewards with half size at 0.25 and 0.75 probability, and rewards of different magnitudes delivered deterministically with 0 or 100% probability.

### Neural recording

After completing behavioral training, each monkey was implanted with two 48- 48-electrode Utah arrays, positioned in the frontal and parietal areas, in the left hemisphere for monkey Mj and the right hemisphere for monkey Mc. In all the arrays, the electrodes were 1.5 mm long, had an impedance of 0.1–0.8 Ohms, and were arranged in rectangular grids with 1-mm spacing (monkey Mj, 7 mm × 7 mm; monkey Mc, 5 mm × 10 mm). The frontal grids were positioned between the anterior bank of the arcuate sulcus and the posterior tip of the principal sulcus. Because of constraints posed by the monkeys' individual vasculature, the frontal array was implanted slightly dorsal to the PS tip in monkey Mj and slightly ventral to it in monkey McD. Both locations were within the pre-arcuate portion of the frontal lobe that is considered a single functional unit[72], and we refer to it here as the LPFC. The parietal grid in each monkey was positioned in the posterior portion of area 7A immediately lateral to the intraparietal sulcus (area OPT).

The electrode signals were referenced to a separate wire positioned over the dura. Data were recorded using the Cereplex System (Blackrock, Salt Lake City, Utah) over 18 sessions spanning 4 months after array implantation in monkey Mj and 12 sessions spanning 2 months after implantation in monkey Mc.

Neuronal spiking from this dataset was analyzed previously, as described in a pre-print[28]. However, that paper is not yet published, and the data are not public. The present paper is the first description of LFP oscillations and TWs from this data set.

### Statistics and reproducibility

Details on statistical analyses and criteria for excluding data points are given below. Data were recorded for as long as the arrays were viable (i.e., provided a sufficiently large signal-to-noise ratio to identify individual cells), which allowed 18 daily behavioral sessions spanning 4 months after array implantation in monkey Mj and 12 sessions spanning 2 months after implantation in monkey Mc. No statistical method was used to predetermine the sample size. All experimental variables were randomized, as described in the *Task* section. The investigators were blinded, as they could not assign an experimental condition to any given electrode.

### Spectral analysis of the LFP

The raw LFP from each electrode and trial were low pass filtered at 200 Hz and notch filtered at 60 Hz to remove line noise. We removed trials with artifacts (e.g., step-like artifacts in LFP traces) by identifying the peak-to-peak amplitude of the broadband LFP trace and removing trials for which this z-transformed measure was more than half a standard deviation away from the mean across all trials. All the neural analyses are based on the remaining trials in which the LFP signals were uncontaminated by artifacts (N trials: LPFC Mc: PR 2852, NPR 1516; LPFC Mj: PR 5165, NPR 2940; PPC Mc: PR 2987, NPR 1580; PPC Mj: PR 2681, NPR 924). The dominant oscillation frequency was calculated based on the multitaper power spectral density estimation[73] implemented with Chronux[74]. Power spectra were computed separately for

each channel and trial, and the background power spectrum was removed through $1/f$ slope estimation[75].

To identify the power of oscillations at different frequencies throughout the task, the spectrograms were computed using the continuous wavelet transform (CWT) with analytic Morse wavelet family scales[76] corresponding to center frequencies of 1–60 Hz. The hyperparameters of the wavelet transform were optimized automatically based on the wavelet's energy spread [gamma = 3, voice per octave = 10, time-bandwidth = 60, and signal sampling rate = 200 Hz]. For follow-up analyses at specific frequencies, we applied a zero-phase-lag Butterworth second-order filter with a bandwidth of 1.5 Hz, which prevents relative phase deformation.

### Characterizing TW

We implemented a version of the TW characterization method from Zhang et al.[17] applied across frequencies and time points. As illustrated in Fig. S1, at frequencies ranging from 2 to 50 Hz, we first applied a zero-phase band-pass filter with a fixed width of ±1.5 Hz. We used this approach to extract the signals at each electrode and then applied the Hilbert transform to the filtered voltage signal, $V(x, y, t)$, recorded by an electrode located at position $x, y$ on the array

$$V(x, y, t) + i\text{Hb}[V(x, y, t)] = A(x, y, t)e^{i\phi(x,y,t)} \quad (1)$$

Here is the Hilbert transform operator, $A(x, y, t)$ is the instantaneous recorded voltage and is the instantaneous phase for the electrode at position $x, y$ at time $t$. Next, to model phase propagation across the simultaneously measured electrodes at each time point, we use a regression model to fit a plane that models the shift across electrodes at different locations $x$ and $y$:

$$\hat{\phi}(x, y, t) = k_x.x + k_y.y + \phi_{\text{ref}}(t). \quad (2)$$

After fitting this model, and are coefficients that reveal the direction of the plane that best describes the phase propagation in the $x$ and $y$ axes. Next, to measure the consistency of wave propagation across electrodes, we calculated the PGD[47]. We compute PGD as the Pearson correlation between the actual phase of the TW at each timepoint $t$ and the predicted phase from the best-fitting planar wave model:

$$\text{PGD}(t) = \frac{\sum_{x,y}((\phi(x,y,t) - \bar{\phi}(t))(\hat{\phi}(x,y,t) - \bar{\hat{\phi}}(t)))}{\sqrt{\sum_{x,y}(\phi(x,y,t) - \bar{\phi}(t))^2 \sum_{x,y}(\hat{\phi}(x,y,t) - \bar{\hat{\phi}}(t))^2}}. \quad (3)$$

Here is the predicted phase of the signal at each location and timepoint, and is the actual phase. Then, using the slope of the fitted plane wave, we identified the propagation direction as below:

$$\text{Propagation direction}(t) = Arctan\left(\frac{k_y}{k_x}\right). \quad (4)$$

Finally, we measured the spatial propagation speed, following the equation for measuring wave speed in physical systems[77] based on the temporal frequency of the traveling waves () and the wave constants (and

$$\nu = \frac{\omega}{\sqrt{k_x^2 + k_y^2}}. \quad (5)$$

### Generalized linear models (GLM) and regression analysis

To determine how PGD correlates with behavioral variables, we constructed individual GLM models to extract coefficients for each variable of interest—PR, expected value (EV), current reward (CR), and cue

and target locations—at each frequency and time-point (Fig. 4D, Fig. S4). To control for potential confounds, we constructed an additional GLM in which we fit the average PGD in the time-frequency region of interest (Fig. 4D) as a function of PR. This model also included as nuisance regressors the licking rate during the pre-cue epoch and the EV of the previous trial (across the 4 arrays: all $F$'s > 5 and all $p$'s < $10^{-5}$; Mc LPFC: $R^2 = 0.062$, Mj LPFC: $R^2 = 0.011$, Mc PPC: $R^2 = 0.074$, Mj PPC: $R^2 = 0.065$).

To determine if the PR was better predicted by TW strength or other physiological indicators, we constructed an additional GLM model with a binomial link function and the following structure:

$$\begin{aligned}\text{Prior reward} = a_0 &+ a_1\text{PGD} + a_2\,\text{TW Direction} + a_3\,\text{TW Speed}\\ &+ a_4\,\text{LFP Power} + a_5\,\text{LFP Bipolar}_{\text{TWparallel}} + a_6\,\text{LFP Bipolar}_{\text{TWorthogonal}}.\end{aligned}$$
$$(6)$$

PR was an indicator variable with a value of 1 if the prior trial was rewarded and 0 otherwise. The dependent variables were measured as described above in the time-frequency region that showed the highest PR effect (Fig. 4D) Direction was coded as the absolute value of the sine of the angular difference between the TW direction on each trial and the dominant axis of wave propagation across trials, allowing us to enter it as a linear (non-circular) regressor in the model. LFP power was the mean power at the peak frequency, calculated using multitaper power spectral density estimation[73]. The LFP Bipolar regressors were constructed by taking the voltage difference between a pair of electrodes that were selected randomly on each trial, with the constraints that the axis connecting the electrodes was parallel to the primary axis of TW propagation direction and that the axis was orthogonal to this axis. These measures do not require circular statistics and provide alternate measures of the gradient of propagating voltage patterns similar to methods used in EEG field analyses (e.g.,[78]). For all GLM model analyses, the regressors were z-scored, allowing us to compare the coefficient magnitudes across regressors. This model produced, for LPFC: $p < 10^{-163}$, $F = 33.8$, $R^2 = 0.02$ and for PPC $F = 7.5$, $R^2 = 0.11$, $p < 10^{-27}$.

In addition, to measure the proportion of the explained variance for each individual predictor in the linear model, we used MATLAB function *fitlm* to calculate for each predictor:

$$\eta^2 = \frac{SS_{\text{effect}}}{SS_{\text{total}}}. \quad (7)$$

is the sum of squares for the effect of interest and is the total sum of squares for all effects, errors, and interactions in the linear model[79,80]. We calculated the variance explained () for each variable and reported in the text the percentage of the total variance explained by the full model that is explained by each variable independently.

To verify the validity of our GLM models, we computed standard model assessment procedures. First, to ensure that the data fit by our model was not collinear, we calculated Variance Inflation Factors (VIF[81]); Large VIF values indicate that an independent predictor is explained by a linear combination of the other predictors. VIF values start at 1, and values above ~5 are considered to indicate a problematically high degree of multicollinearity[82]. Across all our regression models, the VIFs were strictly less than 1.24, indicating that collinearity was not an issue for interpreting our models. We replicated our assessment of regressor collinearity directly using a correlation matrix and verified that there was not a high degree of collinearity in any pair of variables (all correlations < 0.51). Second, to confirm that the assumption of independence was confirmed for our regression models, we ran Durbin–Watson tests for autocorrelations across trials[83]. For all regression models, Durbin–Watson test $p$ values were not significant (all $p$'s > 0.26), indicating that our data fit

conformed to the assumption of independence of residuals required by the GLM.

**Receiver operating characteristic (ROC) analysis of behavioral sensitivity**

We used an ROC analysis to determine the reliability with which the presence of a PR across a group of trials could be inferred from the monkeys' licking behavior. Based on the distributions of licking rates across trials (measured from [0 to -500ms] before cue onset), we constructed an ROC curve. This curve traced, across a range of decision criteria, the frequency of hits (licking rate higher than the criterion and PR delivered) against the frequency of false alarms (licking rate higher than the criterion and PR not delivered). The Area Under the Curve (AUC) was higher than 0.5 in both monkeys (Mc, 0.85; Mj, 0.89], indicating that licking behavior was a reliable indicator of PR. From the ROC curve, we selected the optimal decision criterion for each monkey (the selected threshold where the ROC curve was furthest from the main diagonal). We then defined trials with consistent and inconsistent licking, as explained in the text.

**Reporting summary**

Further information on research design is available in the Nature Portfolio Reporting Summary linked to this article.

## Data availability

The data sets generated during and analyzed during this study are deposited on the Open Science Framework at https://doi.org/10.17605/OSF.IO/ERD8Q. Source data are provided in this paper.

## Code availability

An open-source code repository for all methods is available on GitHub.

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

## Acknowledgements
The work was supported by the Memory and Cognitive Disorder Award from the McKnight Foundation to J.P.G. and a seed grant from the Columbia University Research Initiatives in Science and Engineering to J.P.G. and J.J.

## Author contributions
N.C.F. and J.P.G. designed the experiment. N.C.F. implemented the experiment and collected the data, E.Z. analyzed the data under the supervision of J.P.G. and J.J. E.Z., J.P.G., and J.J. wrote the paper.

## Competing interests
The authors declare no competing interests.
