## [Peer Review File · Nature Communications]

Beta traveling waves in the monkey frontal and parietal areas encode recent reward historyEditorial Note: This manuscript has been previously reviewed at another journal that is not operating a transparent peer review scheme. This document only contains reviewer comments and rebuttal letters for versions considered at *Nature Communications*.

REVIEWER COMMENTS

Reviewer #1 (Remarks to the Author):

The authors claimed to address my concerns about the contribution of licking rate in the pre-cue interval and the prior trial EV in predicting wave strength using a GLM on page 8, paragraph 1, but I do not see where they document non-significant effects of licking rate and prior trial EV on page 8. They simply state it, but don't show statistical results. Maybe, they are somewhere else in the document but I could find them.

Reviewer #2 (Remarks to the Author):

This is the first time I have read this manuscript. I didn't read previous versions. Traveling waves (TWs) have been found in a variety of brain regions recently, and the elucidation of their function in particular could lead to major advances in neuroscience. This paper claims to reveal a new function of TW that encodes reward history in non-human primates who performed a visually guided saccade task. The manuscript is clearly and attractively written, and I read it with great interest. However, the analysis on which it is based seems inadequate in some areas, and it is difficult to accept the claims as they are.

The detection of TWs in the two areas of dl PFC and PPC was performed by circular-linear statistics methods and also carefully confirmed by classical spatial subtraction methods, and the TW phenomenon itself seems to be a solid one. However, the main result that TWs have a higher PGD if the monkeys received a reward on the previous trial is not always clear.

Although Fig. 4A is a clear example that highlights the differences, the overlap in the overall distribution as shown in Fig. 4B is large. The PGD difference (mean), even if significant, would be expected to have a small effect size for previous trial types. At the very least, the effect size should be clearly indicated and discussed on that basis. The GLM analysis that follows is either inadequate or the basis for the conclusions is not clear; GLM analysis has

been used frequently in recent years, even in physiological studies. However, I think researchers recognize that the interpretation of its coefficients is generally very difficult and used in a restrained manner. Even though we can roughly estimate the contribution to the equation, even if it is z-scored, from its magnitude, we can say little about the relationship between the independent variables. From that perspective, the explanations and conclusions in Fig. 4F are not convincing. It may be a good idea to evaluate using product (interaction) measure, relative weights, or other indicators when the independent variables are correlated. In particular, the relationship between TW and LFP oscillations is an important issue, as other reviewers have pointed out, and it would be good to address it exactly.

The subsequent analysis of the relationship between PGD and animals' behavioral sensitivity to the prior reward is also interesting and insightful, but again, the use of GLM seems too audacious. The authors seem to have applied the same GLM model under two conditions (Retrieved/Neglected) and compared the magnitude of the corresponding coefficients, which I do not think is a fair comparison. To begin with, the two conditions are classified by monkeys' behavior, so the number of data (points) seems to be different. I think it would be better to consider the null hypothesis $H_0: \alpha_1 (\text{Retrieved}) = \alpha_1 (\text{Neglected})$ by adding a dummy variable and adding an interaction term to the model. I think it is good that the ANOVA authors were doing afterwards was almost doing that and seeing significant interactions. However, since the same trend was observed for oscillation power, the results may change depending on how the ROI was taken, I am not sure how robust the specificity of the PGD was.

Since the significance of the behavioral sensitivity to statistically irrelevant prior rewards in this task/monkeys is not yet known (which is inevitable, since the behavioral task was not designed to address it), I think it is important to reveal what PGD encodes, but it would be of limited significance in terms of elucidating its function. While the authors' argument about a potential mechanism by which TWs influence behavior was very interesting.

Minor comments:

1) I think the MRI atlas in Fig2A is partially unfolded and difficult to understand; please clarify what the light blue color in the atlas points to. It looks like a sulcus, but if so, the

electrode array was partially implanted across the sulcus. Even if it does not straddle the sulcus, the electrode arrays were not perpendicular to the cortex layers but horizontal near the edge, which makes it difficult to interpret the data. The photos in Fig2E are difficult to see, so it would be better to make it clear in the schematic illustration here.

2) For Fig4B, just to be sure, it would be better to specify that blue is PR and red is PNR. "Monkey Mc" is missing.

3) For Fig4C, how did you define the black dot contours, or at least I think it would be good to write what it indicates in legend.

4) For Fig4D & C, the time course of the PGD modulation is quite different from the time course of the modulation of the coefficients. The starting time for significant coefficient appears much earlier than -1s of cue onset. How are these interpreted?

Reply to Reviewers

Reviewer #1

1. The authors claimed to address my concerns about the contribution of licking rate in the pre-cue interval and the prior trial EV in predicting wave strength using a GLM on page 8, paragraph 1, but I do not see where they document non-significant effects of licking rate and prior trial EV on page 8. They simply state it, but don't show statistical results. Maybe, they are somewhere else in the document but I could find them.

We now added details on the nuisance regressors, including the significance of their coefficients and the percent variance explained by each factor (described in *Methods* and on p. 8/9).

Reviewer #2

1. This is the first time I have read this manuscript. I didn't read previous versions. Traveling waves (TWs) have been found in a variety of brain regions recently, and the elucidation of their function in particular could lead to major advances in neuroscience. This paper claims to reveal a new function of TW that encodes reward history in non-human primates who performed a visually guided saccade task. The manuscript is clearly and attractively written, and I read it with great interest. However, the analysis on which it is based seems inadequate in some areas, and it is difficult to accept the claims as they are.

We thank the reviewer for these supportive comments.

2. The detection of TWs in the two areas of dl PFC and PPC was performed by circular-linear statistics methods and also carefully confirmed by classical spatial subtraction methods, and the TW phenomenon itself seems to be a solid one. However, the main result that TWs have a higher PGD if the monkeys received a reward on the previous trial is not always clear. Although Fig. 4A is a clear example that highlights the differences, the overlap in the overall distribution as shown in Fig. 4B is large. The PGD difference (mean), even if significant, would be expected to have a small effect size for previous trial types. At the very least, the effect size should be clearly indicated and discussed on that basis.

These are good points. With respect to the distributions **Fig. 4B**, please note that in the original version, the error bars showed standard deviations rather than standard errors, which magnified the perception of overlap. We now include a revised version that shows standard errors, which are very small, better explaining why the apparently small shifts of the distributions were highly significant (less than 1% probability of occurring by chance; $p < 0.01$ as stated on p. 8, penultimate paragraph).

We also agree that it is important to have an estimate of effect size and calculated it using d' (the ratio between the difference of the means and the combined standard deviation of the distributions). This information is added to the penultimate paragraph on p. 8.

Finally, we added a measure of effect size to the multivariate GLM models which, unlike the simpler comparisons in **Fig. 4B**, control for confounds of pre-cue licking and prior trial EV. As we now state in the revised section on p. 8/9, analysis of the fraction of variance explained by each regressor shows that PR accounted for 30%-80% of the model-explained variance across the arrays, which was much higher than the other regressors.

3. The GLM analysis that follows is either inadequate or the basis for the conclusions is not clear. GLM analysis has been used frequently in recent years, even in physiological studies. However, I think researchers recognize that the interpretation of its coefficients is generally very difficult and used in a restrained manner. Even though we can roughly estimate the contribution to the equation, even if it is z-scored, from its magnitude, we can say little about the relationship between the independent variables. From that perspective, the explanations and conclusions in Fig. 4F are not convincing. It may be a good idea to evaluate using product (interaction) measure, relative weights or other indicators when the independent variables are correlated. In particular, the relationship between TW and LFP oscillations is an important issue, as other reviewers have pointed out, and it would be good to address it exactly.

As we understand it, the reviewer would like for our paper to more specifically validate that prior reward was related to the strength of TW, as opposed to other potentially correlated variables related to TW or LFP oscillations. We appreciate this concern and, in fact, this is why we used a multivariate GLMs, which distinguish between multiple independent variables by letting them simultaneously ‘compete’ to explain each predictor.

When regressors are allowed to compete in this way and are expressed on a common scale (e.g., by z-scoring), the coefficient magnitudes indicate the relative sensitivity of the dependent variable to each of the competing regressors. Thus, the coefficient magnitudes measure the relative weights that the reviewer requires.

Moreover, a significant coefficient indicates that the regressor captured significant variance that cannot not be explained by the other competing regressors. In the revision, we more fully describe this aspect of our model on p. 10 (2nd paragraph). This better justifies our interpretation that the larger coefficients for PGD in **Fig. 4F** indicate that PGD more strongly predicts prior reward as compared to other features of TW and LFP power.

We also agree with the reviewer’s suggestion to have additional indicators. To this end, we added in the same paragraph information about the percent of variance explained – a new indicator that describes not only the slope of the relationship but the variability that was captured by each term. As expected, this measure followed the pattern of the coefficients, showing that the PGD explained 40-80% of the total model variance, far above the other TW or LFP power metrics.

Finally, we considered the reviewer’s suggestion to evaluate the interaction measure. Because the 6 terms in our original model would generate tens of interactions, we evaluated this question in a simplified model that included only PGD, LFP power and their interaction. The PGD and LFP power had significant coefficients in all the arrays, confirming our conclusion that each measure accounts for independent variance in the model. In contrast, the interaction term was significant in one array but non-significant in the others (all $p > 0.3$). Because the interaction was inconsistent (and because the previous analyses already established the independent contributions of the PGD) we chose to omit this analysis from the revised paper for the sake of simplicity. However, we can include it if the reviewer insists.

We thank the reviewer for these suggestions, and we believe that the revised paper makes a clearer and stronger case that prior reward is most reliably encoded by wave strength rather than other physiological variables.

4. The subsequent analysis of the relationship between PGD and animals' behavioral sensitivity to the prior reward is also interesting and insightful, but again, the use of GLM seems too audacious.

The authors seem to have applied the same GLM model under two conditions (Retrieved/Neglected) and compared the magnitude of the corresponding coefficients, which I do not think is a fair comparison. To begin with, the two conditions are classified by monkeys' behavior, so the number of data (points) seems to be different. I think it would be better to consider the null hypothesis $H_0: a_1(\text{Retrieved}) = a_1(\text{Neglected})$ by adding a dummy variable and adding an interaction term to the model. I think it is good that the ANOVA authors were doing afterwards was almost doing that and seeing significant interactions. However, since the same trend was observed for oscillation power, the results may change depending on how the ROI was taken, I am not sure how robust the specificity of the PGD was. Since the significance of the behavioral sensitivity to statistically irrelevant prior rewards in this task/monkeys is not yet known (which is inevitable, since the behavioral task was not designed to address it), I think it is important to reveal what PGD encodes, but it would be of limited significance in terms of elucidating its function. While the authors' argument about a potential mechanism by which TWs influence behavior was very interesting.

We agree with the reviewer that the definitive analysis for **Fig. 5** is an interaction analysis. We believe that the ANOVA we originally included documented the interaction, as the reviewer notes. Moreover, we believe that the ANOVA is equivalent to the analysis the reviewer suggests – i.e., a GLM that included a dummy variable for behavior (Retrieved/Neglected) and a multiplicative interaction term between PR and behavior.

Nevertheless, we replaced the ANOVA with the GLM analysis the reviewer suggests and lay out the results on p. 10/11. As the reviewer can see, the GLM analysis confirmed the ANOVA results, showing that significant interactions occurred only for TWs in the PFC and not for TW in the PPC or for LFP power in either area. In the revised paragraph (p. 11, top), we also more clearly point out that the analysis in **Fig. S5** was an additional control for potential artefacts of unequal number of trials as it conducted a session level analysis based on a median split (i.e., comparing equal-sized groups with high and low behavioral sensitivity).

Minor comments:

1) I think the MRI atlas in Fig2A is partially unfolded and difficult to understand; please clarify what the light blue color in the atlas points to. It looks like a sulcus, but if so, the electrode array was partially implanted across the sulcus. Even if it does not straddle the sulcus, the electrode arrays were not perpendicular to the cortex layers but horizontal near the edge, which makes it difficult to interpret the data. The photos in Fig2E are difficult to see, so it would be better to make it clear in the schematic illustration here.

In the original **Fig. 2A**, the brain surface was rendered transparently and the blue/gray colors showed deeper landmarks. We agree that that rendition, though esthetically pleasing, was difficult to interpret. We thus replaced it with a rendition showing the visible cortical surface, which is more standard and corresponds better with the photographs in **Fig. 3E**.

2) For Fig4B, just to be sure, it would be better to specify that blue is PR and red is PNR. “Monkey Mc” is missing.

Thanks for catching this. We updated the figure and legend.

3) For Fig4C, how did you define the black dot contours, or at least I think it would be good to write what it indicates in legend.

The contours simply indicate regions defined by pixels that touched each other and had significant coefficients, $p < 0.05$. This is now included in the legend.

4) For Fig4D & C, the time course of the PGD modulation is quite different from the time course of the modulation of the coefficients. The starting time for significant coefficient appears much earlier than -1s of cue onset. How are these interpreted?

The figure was drawn out to the median fixation duration (~1,200 ms). The presence of significant effects at that point is thus explained by the 50% of trials in which the monkeys were already fixating by that point.

REVIEWER COMMENTS

Reviewer #1 (Remarks to the Author):

The authors have addressed all my concerns.

Reply to Authors

Reviewer #2

1. This is the first time I have read this manuscript. I didn't read previous versions. Traveling waves (TWs) have been found in a variety of brain regions recently, and the elucidation of their function in particular could lead to major advances in neuroscience. This paper claims to reveal a new function of TW that encodes reward history in non-human primates who performed a visually guided saccade task. The manuscript is clearly and attractively written, and I read it with great interest. However, the analysis on which it is based seems inadequate in some areas, and it is difficult to accept the claims as they are.

We thank the reviewer for these supportive comments.

I appreciate your agreement with my concerns and thank you for addressing them. Thanks to the additional explanations and clarifications, the authors' intentions are now better understood. However, I am not sure that my concerns have been fully addressed. I am also afraid that the revised version may have gone rather the opposite or too far of what is legitimately claimed, although this may be due in part to my inadequate pointing out for the previous version. I will try to explain my concerns and suggestions more carefully below.

2. The detection of TWs in the two areas of dl PFC and PPC was performed by circular-linear statistics methods and also carefully confirmed by classical spatial subtraction methods, and the TW phenomenon itself seems to be a solid one. However, the main result that TWs have a higher PGD if the monkeys received a reward on the previous trial is not always clear. Although Fig. 4A is a clear example that highlights the differences, the overlap in the overall distribution as shown in Fig. 4B is large. The PGD difference (mean), even if significant, would be expected to have a small effect size for previous trial types. At the very least, the effect size should be clearly indicated and discussed on that basis.

These are good points. With respect to the distributions **Fig. 4B**, please note that in the original version, the error bars showed standard deviations rather than standard errors, which magnified the perception of overlap. We now include a revised version that shows standard errors, which are very small, better explaining why the apparently small shifts of the distributions were highly significant (less than 1% probability of occurring by chance; $p < 0.01$ as stated on p. 8, penultimate paragraph).

I agree that SE is better if you want to show that there is a difference in the means. However, I was not trying to point out the perception of overlap, but rather that there is a lot of overlap in the two distributions and that the subsequent analysis, predicting the prior reward from the TW strength, is expected to be difficult.

We also agree that it is important to have an estimate of effect size and calculated it using d' (the ratio between the difference of the means and the combined standard deviation of the distributions). This information is added to the penultimate paragraph on p. 8.

Thank you for clarifying the effect size. The fact that the effect size is small should be seriously contemplated when considering the function of TW strength, or which neural features are responsible for RH. It seems good that quantitative indicators were given so that we can compare them with data from other studies that have been obtained and will be obtained in the future.

Finally, we added a measure of effect size to the multivariate GLM models which, unlike the simpler comparisons in **Fig. 4B**, control for confounds of pre-cue licking and prior trial EV. As we now state in the revised section on p. 8/9, analysis of the fraction of variance explained by each regressor shows that PR accounted for 30%-80% of the model-explained variance across the arrays, which was much higher than the other regressors.

I understand that you evaluated what PGD encodes using the encoding model with the GLM including the confounds. However, it is difficult to evaluate the analysis of the fraction of variance explained here because the definition is not sufficiently described. Please add a description to the Method.

3. The GLM analysis that follows is either inadequate or the basis for the conclusions is not clear. GLM analysis has been used frequently in recent years, even in physiological studies. However, I think researchers recognize that the interpretation of its coefficients is generally very difficult and used in a restrained manner. Even though we can roughly estimate the contribution to the equation, even if it is z-scored, from its magnitude, we can say little about the relationship between the independent variables. From that perspective, the explanations and conclusions in Fig. 4F are not convincing. It may be a good idea to evaluate using product (interaction) measure, relative weights or other indicators when the independent variables are correlated. In particular, the relationship between TW and LFP oscillations is an important issue, as other reviewers have pointed out, and it would be good to address it exactly.

As we understand it, the reviewer would like for our paper to more specifically validate that prior reward was related to the strength of TW, as opposed to other potentially correlated variables related to TW or LFP oscillations. We appreciate this concern and, in fact, this is why we used a multivariate GLMs, which distinguish between multiple independent variables by letting them simultaneously ‘compete’ to explain each predictor.

I am not an expert on GLM, but as far as I understand, the magnitude of the coefficients of a multivariate GLM can be used as an assessment of the importance of the variable only when the variables are independent of each other, and the model as a whole explains the phenomenon (dependent variable, RH in this case) sufficiently (which consists of certain independent variables). Even if a certain independent variable is the strongest predictor in the model, if the model does not sufficiently explain the phenomenon, it is assumed that there are better models that include other independent variables, and the relationship between PGD and other TW features or LFP oscillations (or coefficients for them in the model) can easily change depending on how much common explanatory power between the additional predictors and PGD or FP features have. I understand that in the case of many physiology data these conditions are unlikely to be fully met, so interpretation is limited in many cases. In the current case, the independent variables are first correlated with each other, and there is no mention of how well the model as a whole explains the independent variables or the sufficiency of the model. Thus, I disagree with the authors' view that “a multivariate GLMs, which distinguish between multiple independent variables by letting them simultaneously ‘compete’ to explain each predictor.”

The analysis of the fraction of variance explained the author added may address this issue, but it is difficult to evaluate the analysis of because the definition is not sufficiently described.

In L228-231 as described, “This confirmed that PGD strength was the strongest predictor of prior trial reward, accounting for 75.8% of the variance explained in the LPFC and 39.8% in the PPC, whereas LFP power accounted for smaller fractions (respectively, 6.8% and 27.2%) and the other metrics accounted for less than 4%”. I assume that “variance explained” here refers to 75.8% of the total model variance explained. Again, the total model variance explained would be a matter and given the low magnitude of the TW strength effect size, I would guess that the sufficiency of the model is not that great. So, as a specific suggestion, it would be better to first state the total variance explained as an indicator of the sufficiency of the model. and add a description for the analysis of the fraction of variance explained in the Method and also clarify the preconditions for the analysis and its appropriate interpretation. In this method is there any procedure to make the regressor uncorrelated?

Not an important issue, but I'm not sure if multivariate GLMs 'compete' to explain each predictor. I feel that sometimes it is better to say that they 'collaborate', in the sense that multivariates increase their predictive power as a model and in the sense that they collaborate with each other. As an aside, I don't mean to recommend using it for this analysis, but I feel that Lasso regression, for example, is closer to the expression of 'compete'.

When regressors are allowed to compete in this way and are expressed on a common scale (e.g., by z-scoring), the coefficient magnitudes indicate the relative sensitivity of the dependent variable to each of the competing regressors. Thus, the coefficient magnitudes measure the relative weights that the reviewer requires.

I disagree on this point, as I said in one previous comment and in my comment for the previous version (the comment was not enough specific and may have been inadequate). I think it is only under certain conditions that weights can be compared, even if they are z-scored.

Moreover, a significant coefficient indicates that the regressor captured significant variance that cannot not be explained by the other competing regressors. In the revision, we more fully describe this aspect of our model on p. 10 (2nd paragraph). This better justifies our interpretation that the larger coefficients for PGD in **Fig. 4F** indicate that PGD more strongly predicts prior reward as compared to other features of TW and LFP power.

I am not so sure that “a significant coefficient indicates that the regressor captured significant variance that cannot not be explained by the other competing regressors.” in general. If two correlated regressors with high but shared explanation power are included in the model, coefficients for both regressors can be significant?

We also agree with the reviewer’s suggestion to have additional indicators. To this end, we added in the same paragraph information about the percent of variance explained – a new indicator that describes not only the slope of the relationship but the variability that was captured by each term. As expected, this measure followed the pattern of the coefficients, showing that the PGD explained 40-80% of the total model variance, far above the other TW or LFP power metrics.

Please add a detailed description of the analysis of the fraction of variance explained in the Methods section. The authors might have their own references, but I found a brief paper regarding the variable importance in multiple linear regression that might be useful to share my concerns and possibly the solution. Interpreting multiple linear regression: A guidebook

of variable importance. Nathans L.L., Oswald F.L., Nimon K.(2012) Practical Assessment, Research and Evaluation, 17 (9) , pp. 1-19.

Finally, we considered the reviewer's suggestion to evaluate the interaction measure. Because the 6 terms in our original model would generate tens of interactions, we evaluated this question in a simplified model that included only PGD, LFP power and their interaction. The PGD and LFP power had significant coefficients in all the arrays, confirming our conclusion that each measure accounts for independent variance in the model. In contrast, the interaction term was significant in one array but non-significant in the others (all $p > 0.3$). Because the interaction was inconsistent (and because the previous analyses already established the independent contributions of the PGD) we chose to omit this analysis from the revised paper for the sake of simplicity. However, we can include it if the reviewer insists.

I think it's fine to describe it, including the fact that it's not reproducible. I think it is fine to describe it, including the fact that it is not always consistent. That being said, and considering the discussion in the comments above, the independence of TW strength and LFP power is certainly important, as another reviewer also pointed out (though he/she seems to be out now). But if the analysis remains as it is, the basis for that independence is, I must say, somewhat tenuous. Of course, further analysis could strengthen it, but I have come to think that it might be better to reserve a conclusion about independence, including the interactions here. The importance of the finding that TW strength encodes RH is unchanged in itself.

As for the relationship between TW strength and LFP power, I feel that the analysis and interpretation of Fig. 3C is somewhat unclear. The fact that the respective peaks coincides is meaningful, but what does it mean that they are correlated? If the slope is less than 1, does it mean that the peaks are displaced? I think it would be better to reconsider this analysis and argue for the possibility of independence of TW strength and LFP power instead of relying solely on regression analysis.

We thank the reviewer for these suggestions, and we believe that the revised paper makes a clearer and stronger case that prior reward is most reliably encoded by wave strength rather than other physiological variables.

I would like to thank the authors for attempting to answer my concern by new analysis. However, while I understand the authors' intentions, I do not believe their methods and interpretations are sufficient.

4. The subsequent analysis of the relationship between PGD and animals' behavioral sensitivity to the prior reward is also interesting and insightful, but again, the use of GLM seems too audacious. The authors seem to have applied the same GLM model under two conditions (Retrieved/Neglected) and compared the magnitude of the corresponding coefficients, which I do not think is a fair comparison. To begin with, the two conditions are classified by monkeys' behavior, so the number of data (points) seems to be different. I think it would be better to consider the null hypothesis $H_0: a_1(\text{Retrieved}) = a_1(\text{Neglected})$ by adding a dummy variable and adding an interaction term to the model. I think it is good that the ANOVA authors were doing afterwards was almost doing that and seeing significant interactions. However, since the same trend was observed for oscillation power, the results may change depending on how the ROI was taken, I am not sure how robust the specificity of the PGD was. Since the significance of the behavioral sensitivity to statistically irrelevant prior rewards in this task/monkeys is not yet known (which is inevitable, since the behavioral task was not designed to address it), I think it is

important to reveal what PGD encodes, but it would be of limited significance in terms of elucidating its function. While the authors' argument about a potential mechanism by which TWs influence behavior was very interesting.

We agree with the reviewer that the definitive analysis for **Fig. 5** is an interaction analysis. We believe that the ANOVA we originally included documented the interaction, as the reviewer notes. Moreover, we believe that the ANOVA is equivalent to the analysis the reviewer suggests – i.e., a GLM that included a dummy variable for behavior (Retrieved/Neglected) and a multiplicative interaction term between PR and behavior.

Nevertheless, we replaced the ANOVA with the GLM analysis the reviewer suggests and lay out the results on p. 10/11. As the reviewer can see, the GLM analysis confirmed the ANOVA results, showing that significant interactions occurred only for TWs in the PFC and not for TW in the PPC or for LFP power in either area. In the revised paragraph (p. 11, top), we also more clearly point out that the analysis in **Fig. S5** was an additional control for potential artefacts of unequal number of trials as it conducted a session level analysis based on a median split (i.e., comparing equal-sized groups with high and low behavioral sensitivity).

I appreciate additional GLM analysis for fair comparison and am glad to see the same result as ANOVA. But the robust or consistency of the result is not so clear, in Fig 5C in Monkey Mc had a difference in PNR condition while Mj had a difference in PNR condition PR condition. What is the implication for the function of TW strength?

Minor comments:

1) I think the MRI atlas in Fig2A is partially unfolded and difficult to understand; please clarify what the light blue color in the atlas points to. It looks like a sulcus, but if so, the electrode array was partially implanted across the sulcus. Even if it does not straddle the sulcus, the electrode arrays were not perpendicular to the cortex layers but horizontal near the edge, which makes it difficult to interpret the data. The photos in Fig2E are difficult to see, so it would be better to make it clear in the schematic illustration here.

In the original **Fig. 2A**, the brain surface was rendered transparently and the blue/gray colors showed deeper landmarks. We agree that that rendition, though esthetically pleasing, was difficult to interpret. We thus replaced it with a rendition showing the visible cortical surface, which is more standard and corresponds better with the photographs in **Fig. 3E**.

I agree that the new version is superior in that it is easier to interpret.

2) For Fig4B, just to be sure, it would be better to specify that blue is PR and red is PNR. “Monkey Mc” is missing.

Thanks for catching this. We updated the figure and legend.

Confirmed.

3) For Fig4C, how did you define the black dot contours, or at least I think it would be good to write what it indicates in legend.

The contours simply indicate regions defined by pixels that touched each other and had significant coefficients, $p < 0.05$. This is now included in the legend.

Thank you very much for clarification.

4) For Fig4D & C, the time course of the PGD modulation is quite different from the time course of the modulation of the coefficients. The starting time for significant coefficient appears much earlier than -1s of cue onset. How are these interpreted?

The figure was drawn out to the median fixation duration (~1,200 ms). The presence of significant effects at that point is thus explained by the 50% of trials in which the monkeys were already fixating by that point.

Thank you very much for clarification. The authors claimed PGD modulation for RH as well as licking behavior emerged after the next trial onset and it is an important observation. However, plots about PGD and licking also show effects from the beginning of the plot. It may be better to have a plot depicting the emerging point of the effect.

We thank Reviewer 2 for additional questions and clarifications. The text below has the entire comment history. The reviewer's first round is in gray, **our first reply s is in blue**, the reviewer's second round is in black, and **our latest reply is in red**. References are listed on the last page. In the **manuscript file, new or changed text** is highlighted yellow.

1. This is the first time I have read this manuscript. I didn't read previous versions. Traveling waves (TWs) have been found in a variety of brain regions recently, and the elucidation of their function in particular could lead to major advances in neuroscience. This paper claims to reveal a new function of TW that encodes reward history in non-human primates who performed a visually guided saccade task. The manuscript is clearly and attractively written, and I read it with great interest. However, the analysis on which it is based seems inadequate in some areas, and it is difficult to accept the claims as they are.

We thank the reviewer for these supportive comments.

I appreciate your agreement with my concerns and thank you for addressing them. Thanks to the additional explanations and clarifications, the authors' intentions are now better understood. However, I am not sure that my concerns have been fully addressed. I am also afraid that the revised version may have gone rather the opposite or too far of what is legitimately claimed, although this may be due in part to my inadequate pointing out for the previous version. I will try to explain my concerns and suggestions more carefully below.

Thank you. We attempted to reply to your comments below as far as we understand them.

2. The detection of TWs in the two areas of dl PFC and PPC was performed by circular-linear statistics methods and also carefully confirmed by classical spatial subtraction methods, and the TW phenomenon itself seems to be a solid one. However, the main result that TWs have a higher PGD if the monkeys received a reward on the previous trial is not always clear. Although Fig. 4A is a clear example that highlights the differences, the overlap in the overall distribution as shown in Fig. 4B is large. The PGD difference (mean), even if significant, would be expected to have a small effect size for previous trial types. At the very least, the effect size should be clearly indicated and discussed on that basis.

These are good points. With respect to the distributions Fig. 4B, please note that in the original version, the error bars showed standard deviations rather than standard errors, which magnified the perception of overlap. We now include a revised version that shows standard errors, which are very small, better explaining why the apparently small shifts of the distributions were highly significant (less than 1% probability of occurring by chance; $p < 0.01$ as stated on p. 8, penultimate paragraph).

I agree that SE is better if you want to show that there is a difference in the means. However, I was not trying to point out the perception of overlap, but rather that there is a lot of overlap in the two distributions and that the subsequent analysis, predicting the prior reward from the TW strength, is expected to be difficult.

Thank you for clarifying.

We also agree that it is important to have an estimate of effect size and calculated it using d' (the ratio between the difference of the means and the combined standard deviation of the distributions). This information is added to the penultimate paragraph on p. 8.

Thank you for clarifying the effect size. The fact that the effect size is small should be seriously contemplated when considering the function of TW strength, or which neural features are responsible for RH. It seems good that quantitative indicators were given so that we can compare them with data from other studies that have been obtained and will be obtained in the future.

We agree that d' provides useful information for comparison with future work.

Finally, we added a measure of effect size to the multivariate GLM models which, unlike the simpler comparisons in Fig. 4B, control for confounds of pre-cue licking and prior trial EV. As we now state in the revised section on p. 8/9, analysis of the fraction of variance explained by each regressor shows that PR accounted for 30%-80% of the model-explained variance across the arrays, which was much higher than the other regressors.

I understand that you evaluated what PGD encodes using the encoding model with the GLM including the confounds. However, it is difficult to evaluate the analysis of the fraction of variance explained here because the definition is not sufficiently described.

We used a standard *Matlab* function to compute the fraction of variance explained. As requested, we now included the specific formula into the *Methods* section on P17/P18.

3. The GLM analysis that follows is either inadequate or the basis for the conclusions is not clear. GLM analysis has been used frequently in recent years, even in physiological studies. However, I think researchers recognize that the interpretation of its coefficients is generally very difficult and used in a restrained manner. Even though we can roughly estimate the contribution to the equation, even if it is z-scored, from its magnitude, we can say little about the relationship between the independent variables. From that perspective, the explanations and conclusions in Fig. 4F are not convincing. It may be a good idea to evaluate using product (interaction) measure, relative weights or other indicators when the independent variables are correlated. In particular, the relationship between TW and LFP oscillations is an important issue, as other reviewers have pointed out, and it would be good to address it exactly.

As we understand it, the reviewer would like for our paper to more specifically validate that prior reward was related to the strength of TW, as opposed to other potentially correlated variables related to TW or LFP oscillations. We appreciate this concern and, in fact, this is why we used a multivariate GLMs, which distinguish between multiple independent variables by letting them simultaneously 'compete' to explain each predictor.

I am not an expert on GLM, but as far as I understand, the magnitude of the coefficients of a multivariate GLM can be used as an assessment of the importance of the variable only when the variables are independent of each other, and the model as a whole explains the phenomenon (dependent variable, RH in this case) sufficiently (which consists of certain independent variables). Even if a certain independent variable is the strongest predictor in the model, if the model does not sufficiently explain the phenomenon, it is assumed that there are better models that include other independent variables, and the relationship between PGD and other TW features or LFP oscillations (or coefficients for them in the model) can easily change depending on how much common explanatory power between the additional predictors and PGD or FP features have. I understand that in the case of many physiology data these conditions are unlikely to be fully met, so interpretation is limited in many cases. In the current case, the independent variables are first correlated with each other, and there is no mention of how well the model as a whole explains the independent variables or the sufficiency of the model. Thus, I disagree with the authors' view that "a multivariate GLMs, which distinguish between multiple independent variables by letting them simultaneously 'compete' to explain each predictor."

The analysis of the fraction of variance explained the author added may address this issue, but it is difficult to evaluate the analysis of because the definition is not sufficiently described.

In L228-231 as described, "This confirmed that PGD strength was the strongest predictor of prior trial reward, accounting for 75.8% of the variance explained in the LPFC and 39.8% in the PPC, whereas LFP power accounted for smaller fractions (respectively, 6.8% and 27.2%) and the other metrics accounted for less than 4%". I assume that "variance explained" here refers to 75.8% of the total model variance

explained. Again, the total model variance explained would be a matter and given the low magnitude of the TW strength effect size, I would guess that the sufficiency of the model is not that great.

So, as a specific suggestion, it would be better to first state the total variance explained as an indicator of the sufficiency of the model. and add a description for the analysis of the fraction of variance explained in the Method and also clarify the preconditions for the analysis and its appropriate interpretation. In this method is there any procedure to make the regressor uncorrelated?

We thank the reviewer for the specific suggestions and we implemented them all in the revised manuscript. As we noted above, we added a detailed description of the method we used to calculate the fraction of variance explained by each predictor. In addition, we included the R^2 values along with the F- and p values for each of our GLM models (P. 11 and P. 17). Finally, we added tests for variance inflation factor and autocorrelations (P. 18) addressing the reviewer's concerns about collinearity, as we explain further below.

The two questions the reviewer repeatedly brings up concern (1) the interpretation of total model R^2 and (2) the role of collinearity (correlations between model regressors). These are fundamental questions in linear regression analyses and we address each in turn.

Model R^2 We agree with the reviewer that, for many neurophysiological studies, "the best" model of a phenomenon is typically unknown and the statistical models that are available have low R^2 indicating that they only partially explain the phenomenon. This is particularly the case in cortical areas like the parietal and frontal cortex which, by virtue of their associative nature, are influenced by multiple factors, only few of which can be measured in any given experiment. The low R^2 values we find are thus consistent with the accepted roles of these areas. Crucially, these values *are fully consistent with the expectations from previous studies* (for a few recent examples, see Knudsen and Wallis, 2020, *Neuron*; Li et al, 2022, *Nature Communications*).

Also crucially for our purpose, it is an accepted statistical principle that *small R^2 values do not preclude interpretation of the relative magnitude of the coefficients*. For example, a statistics textbook (Frost, 2019): states: "*small R-squared values are not always a problem [...] ...if you have a low R-squared value but the independent variables are statistically significant, you can still draw important conclusions about the relationships between the variables...In a nutshell, if your primary goal is to understand the nature of the relationships in your data, a low R-squared is probably not a problem*" (P. 127, 129).

To provide an intuition for this view, we conducted a simple simulation of a linear relationship between two random variables with intercept equal to 0 and slope equal to 0.5 ($Y = 0 + 0.5 \cdot X$), to which we added normally distributed noise to Y and used the *fitlm* function in Matlab (the same function we used to fit our GLM models) to recover the coefficients (the full code is in the box below).

As the reviewer can see, the model has a low R^2 value, similar to those in our paper, reflecting its inability to capture the added random noise. Nevertheless, the model has a high F statistic and very low p-value ($p = 3.7 \times 10^{-6}$) and *correctly* recovers the true parameters (producing an intercept of 0.007 ± 0.06269 that is not statistically different from 0 and a slope that is highly significant and similar to the true value of 0.5 (0.4145 ± 0.08866)). This is a concrete demonstration that, even when the R^2 is low because of unexplained factors or noise, GLM models correctly capture true linear relationships and their specific parameters.

```
N = 250;
x = [zeros(1, N), ones(1, N)];
y = [normrnd(0, 1, 1, N), normrnd(0.5, 1, 1, N)];
L = fitlm(x', y');
disp(L)
```

Linear regression model:				
$y \sim 1 + x1$				
Estimated Coefficients:				
	Estimate	SE	tStat	pValue
(Intercept)	0.0075721	0.062696	0.12077	0.90392
x1	0.41475	0.088666	4.6777	3.7427e-06

Number of observations: 500, Error degrees of freedom: 498
Root Mean Squared Error: 0.991
R-squared: 0.0421, Adjusted R-Squared: 0.0402
F-statistic vs. constant model: 21.9, p-value = 3.74e-06

Collinearity is the second major concern that the reviewer brings up in the comments above, and again in a few other places, e.g.:

Not an important issue, but I'm not sure if multivariate GLMs 'compete' to explain each predictor. I feel that sometimes it is better to say that they 'collaborate', in the sense that multivariates increase their predictive power as a model and in the sense that they collaborate with each other. As an aside, I don't mean to recommend using it for this analysis, but I feel that Lasso regression, for example, is closer to the expression of 'compete'.

When regressors are allowed to compete in this way and are expressed on a common scale (e.g., by z-scoring), the coefficient magnitudes indicate the relative sensitivity of the dependent variable to each of the competing regressors. Thus, the coefficient magnitudes measure the relative weights that the reviewer requires.

I disagree on this point, as I said in one previous comment and in my comment for the previous version (the comment was not enough specific and may have been inadequate). I think it is only under certain conditions that weights can be compared, even if they are z-scored.

Moreover, a significant coefficient indicates that the regressor captured significant variance that cannot not be explained by the other competing regressors. In the revision, we more fully describe this aspect of our model on p. 10 (2nd paragraph). This better justifies our interpretation that the larger coefficients for PGD in Fig. 4F indicate that PGD more strongly predicts prior reward as compared to other features of TW and LFP power.

I am not so sure that “a significant coefficient indicates that the regressor captured significant variance that cannot not be explained by the other competing regressors.” in general. If two correlated regressors with high but shared explanation power are included in the model, coefficients for both regressors can be significant?

The question of correlated regressors – collinearity – is at the heart of *all* linear regression analyses, including GLM, which are designed to disentangle contributions of different regressors to a single dependent variable. The reviewer is correct that these methods work best when the regressors are perfectly uncorrelated ($r = 0$), because correlations greater than 0 (collinearity) make it more difficult to assign credit to one variable or the other. However, what makes multiple regression so extremely useful is that the scale of the problem has a very non-linear dependence on the strength of the correlations. Because of this nonlinearity, multivariate regression is *the method of choice* for a range of practical applications in which the correlations are in the low to moderate range, as was the case in our paper.

This point is best illustrated by the plot to the right, which we reproduced from a classic text on statistics (*Statistical Rethinking*, McElreath, 2016, p. 149).

FIGURE 5.10. The effect of correlated predictor variables on the narrowness of the posterior distribution. The vertical axis shows the standard deviation of the posterior distribution of the slope. The horizontal axis is the correlation between the predictor of interest and another predictor added to the model. As the correlation increases, the standard deviation inflates.

In this plot, the y axis is the standard deviation of the posterior estimate over coefficients (which is identical to the standard error for OLS regression) in a scenario in which we have enough data to estimate each parameter to a confidence of 0.001 for uncorrelated predictors ($r = 0$ on the x axis). Correlations have little effect until a value of 0.6 - 0.8, and produce a dramatic rise in the standard deviation only after this point. (In Bayesian terms, the rise at high correlation values means that the posterior distribution over the coefficient estimates gets narrower more slowly than the ideal $\text{sq-rt}(N)$ factor that we would expect with perfectly orthogonal variables. In frequentist terms, it means that correlations above ~ 0.6 -0.8 dramatically reduce statistical power – i.e., require more and more data to detect the true effect.)

Because of this fundamental dependence, correlations below 0.6-0.8 are considered appropriate for multiple regression analysis (with most authors adopting a cutoff of 0.7). This is a universal practice that can be documented with thousands of references from multiple domains (e.g., see Dormann et al. 2006; Heikkinen et al., 2006)

As we now clearly state in the text (*Methods*, P. 18), the correlations in our data are well below this critical range (r 's = 0.18, 0.35, 0.48, 0.51). Thus, our model results are similar to what they would be if we had perfectly uncorrelated regressors (i.e., close to $r = 0$ in the above plot).

To complement the reasoning above, we conducted two additional tests that are classic measures of collinearity: the Variance Inflation Factor (VIF) and Durbin-Watson test for autocorrelations.

VIF measures the degree of multicollinearity for each model regressor. According to Becker et al (2015) VIF should be less than 5 or 10 for GLM modeling, although a stricter threshold of 2.5 is sometimes used. As we now report on **P. 18** of the main text, the VIF values across all of our models were well below this cutoff (all VIF < 1.24). In the table below we provide more detail about all the individual values for the model in **Fig. 4F** the reviewer specifically addresses. Thus, consistent with the low correlation coefficients, all the VIF values are in acceptable ranges.

	PGD	Direction	Speed	Power	Bipolar (parallel)	Bipolar (Orthogonal)
LPFC	1.0702	1.0010	1.0013	1.0479	1.2200	1.2449
PPC	1.0982	1.0035	1.0083	1.0940	1.1057	1.1083

A second concern is that we may have had autocorrelations between successive trials in the data – e.g., the PGD on trial 2 may have depended on that in trial 1, etc. To rule out this concern, we conducted a Durbin–Watson test to check for autocorrelation in the residuals of the regression model (Durbin et al, 1950). As we note on **P. 18**, none of the tests were significant, providing no evidence that autocorrelations in the data violated the assumptions of the GLM analyses.

Together, these considerations mean that the multivariate approach we adopted is fully appropriate for our data. Our results are similar to that they would be for perfectly uncorrelated regressors. We can also be confident that the regressors compete not “collaborate” with each other, as the reviewer mentions (“collaboration” happens at correlation values of $r > 0.8$, when the large standard deviations of the estimates may cause credit to be inappropriately allocated to highly correlated predictors or to the model as a whole). In sum, we can consider the significance, relative magnitudes, and % variance explained as indicators of the credit that is correctly assigned to each regressor, as we state in the text.

We also agree with the reviewer’s suggestion to have additional indicators. To this end, we added in the same paragraph information about the percent of variance explained – a new indicator that describes not only the slope of the relationship but the variability that was captured by each term. As expected, this measure followed the pattern of the coefficients, showing that the PGD explained 40-80% of the total model variance, far above the other TW or LFP power metrics.

Please add a detailed description of the analysis of the fraction of variance explained in the Methods section. The authors might have their own references, but I found a brief paper regarding the variable importance in multiple linear regression that might be useful to share my concerns and possibly the solution. Interpreting multiple linear regression: A guidebook of variable importance. Nathans L.L., Oswald F.L., Nimon K.(2012) Practical Assessment, Research and Evaluation, 17 (9) , pp. 1-19.

We thank the reviewer for this reference. Unfortunately, we could not find in it a description of % variance explained calculation. However, this method is standard and we not described it as noted above.

Finally, we considered the reviewer’s suggestion to evaluate the interaction measure. Because the 6 terms in our original model would generate tens of interactions, we evaluated this question in a simplified model that included only PGD, LFP power and their interaction. The PGD and LFP power had significant coefficients in all the arrays, confirming our conclusion that each measure accounts for independent variance in the model. In contrast, the interaction term was significant in one array but non-significant in the others (all $p > 0.3$). Because the interaction was inconsistent (and because the previous analyses already established the independent contributions of the PGD) we chose to omit this analysis from the revised paper for the sake of simplicity. However, we can include it if the reviewer insists.

I think it's fine to describe it, including the fact that it's not reproducible. I think it is fine to describe it, including the fact that it is not always consistent. That being said, and considering the discussion in the comments above, the independence of TW strength and LFP power is certainly important, as another reviewer also pointed out (though he/she seems to be out now). But if the analysis remains as it is, the basis for that independence is, I must say, somewhat tenuous. Of course, further analysis could strengthen it, but I have come to think that it might be better to reserve a conclusion about independence, including the interactions here. The importance of the finding that TW strength encodes RH is unchanged in itself.

As for the relationship between TW strength and LFP power, I feel that the analysis and interpretation of Fig. 3C is somewhat unclear. The fact that the respective peaks coincides is meaningful, but what does it mean that they are correlated? If the slope is less than 1, does it mean that the peaks are displaced? I think it would be better to reconsider this analysis and argue for the possibility of independence of TW strength and LFP power instead of relying solely on regression analysis.

As suggested, in the revised paper we now acknowledge the possibility that the power and TW strength of these oscillations may reflect different signals. To explain both sides of this issue fully, in the revised *Discussion* we have improved our explanation of why power and TW strength may in fact reflect the same oscillation, even as the power analysis apparently identified a slower frequency band (P. 13, last paragraph). We explain that there is a known measurement issue related to the $1/f$ power spectrum of LFP signals, where the frequencies of oscillations identified in peak-picking procedures could be biased to appear at slower frequencies due to the $1/f$ power spectrum shape. We now address this issue in the revised text where we state: “Although **Figure 3C** suggests that the frequencies of power peaks were lower than the frequencies of peak TW strength, this may

have been caused by the known 1/f distribution of LFP power (He, 2014). Due to the 1/f ‘tilt’ in the LFP power spectrum, it would systematically bias our measurements of power peaks to occur at lower frequencies (Donoghue et al, 2020a, Donoghue et al., 2020b), without affecting the measure of peak TW frequency.”

In **Fig. 1** of the paper, we discussed how, even at the same frequency, power and TW strength can vary independently from each other. In panel **A** in the figure below, we reproduce the crux of this argument. We show that, for an oscillation of constant frequency, TW arise only if the phases of waves are spatially organized across electrodes (left column). If the phases vary randomly, LFP power will not produce TW (right column), even if LFP power is very high (bottom right). From the reviewer’s comments above, we were not entirely certain if the reviewer asked for additional clarification on this point. To address this possibility, we added a brief sentence to the *Results* section (**P. 7**, top).

In addition, we included a brief network simulation for the reviewer’s benefit, which we can also include in the paper if the reviewer requests. Following methods from the Sejnowski lab (Moldakarimov et al., 2018, Davis et al., 2021), we simulated two networks of neurons which were, respectively, connected with randomly distributed weights (panel **C**, left) or with weights that had a directional gradient (panel **C**,; see figure caption for additional detail). Both networks generated LFP oscillations (panel **B**) that had with identical spectral composition (panel **D**). However, only the directed network generated strong TW (panel **E**). Thus, a realistic network that has spatially organized connectivity generates TW independently of LFP power.

Computational model illustrating that the topography of a network can control whether an oscillation at a fixed power exhibits a strong traveling wave. A) Illustrations showing possible neural signals with different combinations of high and low LFP power and Traveling Wave strength. LFP power, as visualized along rows, reflects the amplitude of an underlying oscillation. In contrast, Traveling Wave strength, as visualized along the columns, is informative about the timing of the phase of oscillations between nearby electrodes. (B-E) Results of two simulations explaining how TW strength may reflect the spatial structure of the neural network that is generating a given LFP oscillation. B) Generation of a periodic field potential from a population of interconnected Leaky Integrate and Fire (LIF) Neurons (Davis and Sejnowski, 2021). 100 LIF neurons are

connected through synaptic weights coming from a gaussian distribution centered around zero. Injection of a transient input to the network generates a locally stable oscillation. C) Spatial organization of the oscillation generated by this network reflects synaptic topography. The images show the field potentials generated by this network on particular trials when the model was run for two spatial arrangements of synaptic weights: randomly organized (left panel) and the other with topographically arranged synapses (right panel; higher probability for local connections plus 10% strong long-distance connections; Sejnowski et al. 2018, PNAS, Litwan Kumar et al. 2012, Nature Neuroscience). Note that the random network produces a wave with no spatial structure whereas the network with topographically arranged synapses shows a smoothly propagating Traveling Wave. D) Mean LFP power spectra produced by random and topographic networks. Note that LFP power is similar for both models, indicating that LFP power in this type of model does not vary with the spatial structure of synapses. E) Mean TW strength (PGD) from the LFP of random (black) and topographic (red) networks. The topographic network shows a significantly higher PGD compared to the non-topographic network ($p < 0.05$), demonstrating the ability of the PGD measure to effectively measure the strength of continuously varying traveling waves.

We thank the reviewer for these suggestions, and we believe that the revised paper makes a clearer and stronger case that prior reward is most reliably encoded by wave strength rather than other physiological variables.

I would like to thank the authors for attempting to answer my concern by new analysis. However, while I understand the authors' intentions, I do not believe their methods and interpretations are sufficient.

We hope that with these additional revisions we have addressed the reviewer's remaining concerns.

4. The subsequent analysis of the relationship between PGD and animals' behavioral sensitivity to the prior reward is also interesting and insightful, but again, the use of GLM seems too audacious. The authors seem to have applied the same GLM model under two conditions (Retrieved/Neglected) and compared the magnitude of the corresponding coefficients, which I do not think is a fair comparison. To begin with, the two conditions are classified by monkeys' behavior, so the number of data (points) seems to be different. I think it would be better to consider the null hypothesis $H_0: a_1(\text{Retrieved}) = a_1(\text{Neglected})$ by adding a dummy variable and adding an interaction term to the model. I think it is good that the ANOVA authors were doing afterwards was almost doing that and seeing significant interactions. However, since the same trend was observed for oscillation power, the results may change depending on how the ROI was taken, I am not sure how robust the specificity of the PGD was. Since the significance of the behavioral sensitivity to statistically irrelevant prior rewards in this task/monkeys is not yet known (which is inevitable, since the behavioral task was not designed to address it), I think it is important to reveal what PGD encodes, but it would be of limited significance in terms of elucidating its function. While the authors' argument about a potential mechanism by which TWs influence behavior was very interesting.

We agree with the reviewer that the definitive analysis for **Fig. 5** is an interaction analysis. We believe that the ANOVA we originally included documented the interaction, as the reviewer notes. Moreover, we believe that the ANOVA is equivalent to the analysis the reviewer suggests – i.e., a GLM that included a dummy variable for behavior (Retrieved/Neglected) and a multiplicative interaction term between PR and behavior. Nevertheless, we replaced the ANOVA with the GLM analysis the reviewer suggests and lay out the results on p. 10/11. As the reviewer can see, the GLM analysis confirmed the ANOVA results, showing that significant interactions occurred only for TWs in the PFC and not for TW in the PPC or for LFP power in either area. In the revised paragraph (p. 11, top), we also more clearly point out that the analysis in **Fig. S5** was an additional control for potential artefacts of unequal number of trials as it conducted a session level analysis based on a median split (i.e., comparing equal-sized groups with high and low behavioral sensitivity).

I appreciate additional GLM analysis for fair comparison and am glad to see the same result as ANOVA. But the robust or consistency of the result is not so clear, in Fig 5C in Monkey Mc had a difference in PNR

condition while Mj had a difference in PNR condition PR condition. What is the implication for the function of TW strength?

The fact that TW's modulation with behavior differs between monkeys is interesting and may indicate that TW strength reflects the monkeys' individual differences in interpreting the task or attending to different attributes. This possibility can be tested in future research. We now discuss this issue and incorporated our interpretation of this result in the discussion (P. 12).

Minor comments:

1) I think the MRI atlas in Fig2A is partially unfolded and difficult to understand; please clarify what the light blue color in the atlas points to. It looks like a sulcus, but if so, the electrode array was partially implanted across the sulcus. Even if it does not straddle the sulcus, the electrode arrays were not perpendicular to the cortex layers but horizontal near the edge, which makes it difficult to interpret the data. The photos in Fig2E are difficult to see, so it would be better to make it clear in the schematic illustration here.

In the original Fig. 2A, the brain surface was rendered transparently and the blue/gray colors showed deeper landmarks. We agree that that rendition, though esthetically pleasing, was difficult to interpret. We thus replaced it with a rendition showing the visible cortical surface, which is more standard and corresponds better with the photographs in Fig. 3E.

I agree that the new version is superior in that it is easier to interpret.

We are happy this is resolved.

2) For Fig4B, just to be sure, it would be better to specify that blue is PR and red is PNR. "Monkey Mc" is missing.

Thanks for catching this. We updated the figure and legend.

Confirmed.

We are happy this is resolved.

3) For Fig4C, how did you define the black dot contours, or at least I think it would be good to write what it indicates in legend.

The contours simply indicate regions defined by pixels that touched each other and had significant coefficients, $p < 0.05$. This is now included in the legend.

Thank you very much for clarification.

We are happy this is resolved.

4) For Fig4D & C, the time course of the PGD modulation is quite different from the time course of the modulation of the coefficients. The starting time for significant coefficient appears much earlier than -1s of cue onset. How are these interpreted?

The figure was drawn out to the median fixation duration (~1,200 ms). The presence of significant effects at that point is thus explained by the 50% of trials in which the monkeys were already fixating by that point.

Thank you very much for clarification. The authors claimed PGD modulation for RH as well as licking behavior emerged after the next trial onset and it is an important observation. However, plots about PGD and licking also show effects from the beginning of the plot.

Our specific claim is that the reward history effect was not a mere continuation of a response to rewards on the previous trial. This follows from the fact that there were *no* PGD modulations in response to the reward outcome just after reward release (**Fig S4**).

The reviewer asks us to additionally pinpoint the *precise onset times* of the PGD modulation but this is, unfortunately, not feasible in our data. This analysis would require us to align the data on the onset of central fixation on each trial, and inspect a few hundred milliseconds before and after that point. However, our highly trained monkeys often anticipated the trial start and were already fixating by the time we opened the window for data collection on each trial. Any latency analysis we attempted would thus be restricted to less than half of the trials, making the estimates unreliable.

Thus, we must leave the question of onset times for a future project. Here, we conclude confidently that the reward history response was not a continuation to a response to the prior reward, and therefore must have arisen *de novo* at or before the start of the following trial

References:

- [1] Becker, J. M., Ringle, C. M., Sarstedt, M., & Völckner, F. (2015). How collinearity affects mixture regression results. *Marketing Letters*, 26, 643-659.
- [2] Davis, Z. W., Benigno, G. B., Fletteman, C., Desbordes, T., Steward, C., Sejnowski, T. J., ... & Muller, L. (2021). Spontaneous traveling waves naturally emerge from horizontal fiber time delays and travel through locally asynchronous-irregular states. *Nature Communications*, 12(1), 6057.
- [3] Dobson, A. J., & Barnett, A. G. (2018). *An introduction to generalized linear models*. CRC press.
- [4] Dormann, C. F., Elith, J., Bacher, S., Buchmann, C., Carl, G., Carré, G., ... & Lautenbach, S. (2013). Collinearity: a review of methods to deal with it and a simulation study evaluating their performance. *Ecography*, 36(1), 27-46.
- [4] Donoghue, T., Haller, M., Peterson, E. J., Varma, P., Sebastian, P., Gao, R., ... & Voytek, B. (2020). Parameterizing neural power spectra into periodic and aperiodic components. *Nature neuroscience*, 23(12), 1655-1665.
- [5] Donoghue, T., Schaworonkow, N., & Voytek, B. (2022). Methodological considerations for studying neural oscillations. *European journal of neuroscience*, 55(11-12), 3502-3527.
- [6] Durbin, J., and G. S. Watson. "Testing for Serial Correlation in Least Squares Regression I." *Biometrika* 37, pp. 409–428, 1950.
- [7] Frost, J. (2019). *Regression analysis: An intuitive guide for using and interpreting linear models*. Statistics By Jim Publishing.
- [8] Heikkinen, R. K., Luoto, M., Araújo, M. B., Virkkala, R., Thuiller, W., & Sykes, M. T. (2006). Methods and uncertainties in bioclimatic envelope modelling under climate change. *Progress in Physical Geography*, 30(6), 751-777.
- [9] Knudsen, E. B., & Wallis, J. D. (2020). Closed-loop theta stimulation in the orbitofrontal cortex prevents reward-based learning. *Neuron*, 106(3), 537-547.
- [10] Li, Y., Daddaoua, N., Horan, M., Foley, N. C., & Gottlieb, J. (2022). Uncertainty modulates visual maps during noninstrumental information demand. *Nature Communications*, 13(1), 5911.
- [11] Litwin-Kumar, A., & Doiron, B. (2012). Slow dynamics and high variability in balanced cortical networks with clustered connections. *Nature neuroscience*, 15(11), 1498-1505.
- [12] McCullagh, P., & Nelder, J. A. (1989). *Generalized linear models*. CRC press.
- [13] McElreath, R. (2020). *Statistical rethinking: A Bayesian course with examples in R and Stan*. CRC press.
- [14] Moldakarimov, S., Bazhenov, M., Feldman, D. E., & Sejnowski, T. J. (2018). Structured networks support sparse traveling waves in rodent somatosensory cortex. *Proceedings of the National Academy of Sciences*, 115(20), 5277-5282.